 eLife

# Adult-born neurons facilitate olfactory bulb pattern separation during task engagement

Wankun L Li[1,2], Monica W Chu[1,2], An Wu[1,2], Yusuke Suzuki[3], Itaru Imayoshi[4]*, Takaki Komiyama[1,2]*

[1]Neurobiology Section, Center for Neural Circuits and Behavior, University of California, San Diego, San Diego, United States; [2]Department of Neurosciences, University of California, San Diego, San Diego, United States; [3]Medical Innovation Center/SK Project, Graduate School of Medicine, Kyoto University, Kyoto, Japan; [4]Graduate School of Biostudies, Kyoto University, Kyoto, Japan

**Abstract** The rodent olfactory bulb incorporates thousands of newly generated inhibitory neurons daily throughout adulthood, but the role of adult neurogenesis in olfactory processing is not fully understood. Here we adopted a genetic method to inducibly suppress adult neurogenesis and investigated its effect on behavior and bulbar activity. Mice without young adult-born neurons (ABNs) showed normal ability in discriminating very different odorants but were impaired in fine discrimination. Furthermore, two-photon calcium imaging of mitral cells (MCs) revealed that the ensemble odor representations of similar odorants were more ambiguous in the ablation animals. This increased ambiguity was primarily due to a decrease in MC suppressive responses. Intriguingly, these deficits in MC encoding were only observed during task engagement but not passive exposure. Our results indicate that young olfactory ABNs are essential for the enhancement of MC pattern separation in a task engagement-dependent manner, potentially functioning as a gateway for top-down modulation.
DOI: https://doi.org/10.7554/eLife.33006.001

*For correspondence:
iimayosh@virus.kyoto-u.ac.jp (II);
tkomiyama@ucsd.edu (TK)

**Competing interest:** The authors declare that no competing interests exist.

## Introduction

The brain remains plastic throughout life. A dramatic example of neural circuit plasticity during adulthood comes in the form of adult neurogenesis (*Altman and Das, 1965*). The subventricular zone (SVZ) is one of the two main loci in the rodent brain where adult neurogenesis takes place (*Zhao et al., 2008*). In the SVZ, many thousands of new neurons are produced each day throughout adulthood, and these new neurons migrate through the rostral migratory stream to the olfactory bulb (*Lois and Alvarez-Buylla, 1994*), the first olfactory center of the brain. Once in the olfactory bulb, about 95% of the adult-born neurons (ABNs) differentiate into granule cells (GCs) and the majority of the remaining differentiate into periglomerular cells (*Lledo et al., 2006*), both of which are GABAergic local inhibitory neurons. GCs inhibit mitral cells (MCs), the principal neurons of the olfactory bulb, through their dendrodendritic reciprocal connections (*Isaacson and Strowbridge, 1998*; *Rall et al., 1966*; *Shepherd, 1963*). This inhibition of MCs can sparsen odor representations and enhance the signal-to-noise ratio (*Koulakov and Rinberg, 2011*; *Yokoi et al., 1995*; *Yu et al., 2014*). Consistent with this idea, general activation of bulbar inhibitory neurons can accelerate learning (*Abraham et al., 2010*), while suppression of inhibitory neuron activity can increase the excitability of MCs and reduce MC pattern separation (*Gschwend et al., 2015*). Thus, local inhibitory neurons in the olfactory bulb, including ABNs, likely control olfactory perception by providing inhibition onto MCs.

**eLife digest** Most brain cells or neurons form early in life. Yet, in some parts of the brain, new neurons develop throughout adulthood, in a process called adult neurogenesis. These new neurons are incorporated into existing brain circuits and likely help the brain process information. In rodents, adult neurogenesis produces many new cells in the olfactory bulb, a part of the brain that processes smells. This is likely because the sense of smell is important for the survival of these animals.

What these adult-born neurons do and how they aid the rodent's sense of smell is not clear. Previous studies have had conflicting results about whether these cells help animals distinguish smells and under what circumstances. More studies about how these adult-born neurons become incorporated in the brain and how they aid creatures' sense of smell could help scientists studying brain diseases.

Now, Li et al. show that mice that lack adult-born neurons have a difficult time distinguishing very similar smells. In the experiments, mice were genetically engineered to suppress the formation of new neurons in adult animals. These mice lacking adult-born neurons and typical mice were trained to do tasks that require them to distinguish similar or very different scents. While the animals were completing these tasks, Li et al. used a technique called two-photon calcium imaging to see what was happening in cells in the olfactory bulb.

The experiments revealed altered neuron activity in the genetically engineered animals compared with normal ones when they were trying to distinguish similar smells. Yet there was no difference when the mice distinguished very different scents. This suggests that adult-born cells are important for mice working to distinguish scents. The mechanisms at work in the mice may be the same ones that help humans distinguish wines or perfumes. Learning more about how new cells form in adult brains could help scientists understand these processes and develop treatments for brain diseases in humans.

DOI: https://doi.org/10.7554/eLife.33006.016

As ABNs integrate into local circuits, they display higher levels of morphological and functional plasticity during the first ~8 weeks after their birth compared to their later mature stage (*Kelsch et al., 2009*; *Mizrahi, 2007*; *Nissant et al., 2009*; *Sailor et al., 2016*). Furthermore, the spine dynamics, synaptic plasticity, sensory response pattern, as well as survival rate of ABNs are influenced by olfactory experience during this early period (*Alonso et al., 2006*; *Lemasson et al., 2005*; *Lepousez et al., 2014*; *Livneh et al., 2014*; *Mouret et al., 2008*; *Petreanu and Alvarez-Buylla, 2002*; *Quast et al., 2017*; *Rochefort et al., 2002*; *Yamaguchi and Mori, 2005*). These unique and plastic features of young ABNs make it likely that they play a unique role in the processing of the complex and dynamic olfactory environment. Indeed, some studies have shown that ABNs are essential for certain olfactory behaviors such as odor discrimination and association reversal learning (*Alonso et al., 2012*; *Bath et al., 2008*; *Enwere et al., 2004*; *Gheusi et al., 2000*; *Moreno et al., 2009*; *Sakamoto et al., 2014*). However, other studies have found little effects of ABN manipulation on odor discrimination (*Breton-Provencher et al., 2009*; *Imayoshi et al., 2008*; *Lazarini et al., 2009*). Thus, ABNs are not essential for all olfactory processing. Instead, the inconsistencies between these results raise the possibility that the impact of ABNs may depend on the behavioral context.

Consistent with the idea that the functions of ABNs are context-dependent, local inhibitory neurons in the olfactory bulb including ABNs receive abundant glutamatergic centrifugal inputs from higher brain areas such as anterior olfactory nucleus, piriform cortex and entorhinal cortex (*Balu et al., 2007*; *Boyd et al., 2015, 2012*; *Chapuis et al., 2013*; *Kiselycznyk et al., 2006*; *Markopoulos et al., 2012*; *Nunez-Parra et al., 2013*; *Otazu et al., 2015*; *Rothermel et al., 2014*). Inhibitory neurons also express a variety of neuromodulator receptors, which provide additional avenues for top-down modulation (*Castillo et al., 1999*; *Devore and Linster, 2012*; *Ma and Luo, 2012*; *Moreno et al., 2012*; *Rothermel et al., 2014*; *Shipley et al., 1985*). Furthermore, anesthesia inactivates feedback fibers and reduces GC activity while increasing MC activity (*Kato et al., 2012*; *Rothermel and Wachowiak, 2014*). Thus, bulbar inhibitory neurons may function as a mediator of feedback regulation to shape MC odor encoding (*Markopoulos et al., 2012*). Consistent with this notion, it has been reported that cortical feedback can decorrelate MC odor representations (*Otazu et al., 2015*). Similarly, neuromodulatory projections can modulate MC activity (*Rothermel and Wachowiak, 2014*) and improve

perceptual learning (*Ma and Luo, 2012*). However, the role of young ABNs in mediating context-dependent modulation of MC activity has not been fully explored.

To assess the role of ABNs in the olfactory bulb, we adopted a genetic method to inducibly suppress adult neurogenesis. We reasoned that this chronic ablation allows us to probe for the functions of ABNs which cannot be compensated for by other inhibitory neurons. Behavioral experiments showed that ABN ablation animals were impaired in fine, but not coarse, odor discrimination. Neither suppression of hippocampal ABNs alone nor non-selective ablation of a comparable number of bulbar neurons caused the same behavioral deficit, highlighting the unique importance of young ABNs. Two-photon calcium imaging revealed that the behavioral deficit was accompanied by a decreased separation of MC ensemble responses to similar odorants in ablation animals. This decreased separation was largely due to a reduction in suppressive odorant responses by MCs. Interestingly, this difference in suppressive responses was only observed when animals were actively engaged in the task.

## Results

### Ablation of adult-born neurons

To investigate the role of adult neurogenesis in olfactory processing, we adopted the transgenic mouse line *Gfap-tk* (*Snyder et al., 2011*). In this line, herpes simplex virus Thymidine Kinase (TK) is expressed under the Glial Fibrillary Acidic Protein (GFAP) promoter, rendering mitotic neural stem cells sensitive to the antiviral drug Valganciclovir (VGCC) (*Figure 1A*). This gives us a means to specifically suppress adult neurogenesis in an inducible manner.

We sought to investigate the consequences of eliminating young ABNs that are 8 weeks old and younger. To achieve this, we treated *Gfap-tk* mice with VGCC, starting 8 weeks prior to the beginning of the behavioral experiments and continuing throughout the duration of the experiments (*Figure 1B*). Hereafter we refer to these animals as the 'ablation' animals. The ablation of adult neurogenesis in the olfactory bulb was nearly complete, shown by post-hoc BrdU labeling (p<0.0001, Wilcoxon rank sum test, *Gfap-tk*$^+$ vs. *Gfap-tk*$^-$; *Figure 1C*) as well as immunostaining for Doublecortin, a marker for immature neurons (*Figure 1D*). Importantly, there was no difference in the density of GFAP-positive astrocytes in VGCC-treated *Gfap-tk*$^+$ and *Gfap-tk*$^-$ mice in the olfactory bulb (p=0.9911, Wilcoxon rank sum test; *Figure 1E*) nor in the hippocampus dentate gyrus (p=0.1359, Wilcoxon rank sum test), consistent with a previous report (*Snyder et al., 2011*). Another previous report also showed that regeneration of olfactory sensory neurons is intact in this mouse line (*Cummings et al., 2014*). Ablation mice showed no obvious general health impairment, and demonstrated mobility and anxiety levels comparable to control mice in an open field test (average speed: p=1.0000, center time fraction: p=0.7984, Wilcoxon rank sum test; *Figure 1F*).

### ABN ablation impairs fine odor discrimination

Equipped with this effective and specific method of inducible adult neurogenesis ablation, we explored how the absence of young ABNs could affect olfactory behavior. We compared the behavioral performance of the ablation animals and littermate controls (control: n = 22; ablation: n = 23). Both groups were treated identically including VGCC administration, and experimenters were blind to their genotypes during experiments. Mice were trained in a two-alternative-choice olfactory discrimination task under head-fixation. In this task, a certain odorant was delivered in each trial for 4 s, followed by an answer period of 2 s during which mice were required to lick -either the left or right port according to the odorant cue to receive a water reward (*Figure 2A*). Mice were trained daily, one session per day, and each session consisted of 144.4 ± 17.4 trials for the control group and 146.4 ± 13.8 trials for the ablation group.

After the initial pre-training period (Materials and methods), mice were trained in a relatively easy discrimination task in which mice were required to discriminate between conspicuously different binary mixtures (Left lick: 80% heptanal and 20% ethyl-tiglate (80H20E); right lick: 20% heptanal and 80% ethyl-tiglate (20H80E), all mixture percentages are of a total concentration of 100 ppm; *Figure 2B*). Both ablation and control animals achieved expertise in this task (defined as >80% success rate) in equivalent durations of training (number of sessions, control: 4.18 ± 0.21, ablation: 4.39 ± 0.25, mean ±S.E.M; p=0.4523, Wilcoxon rank sum test; *Figure 2C*). Thus, we conclude that young ABNs are not required for the performance of this easy discrimination task.

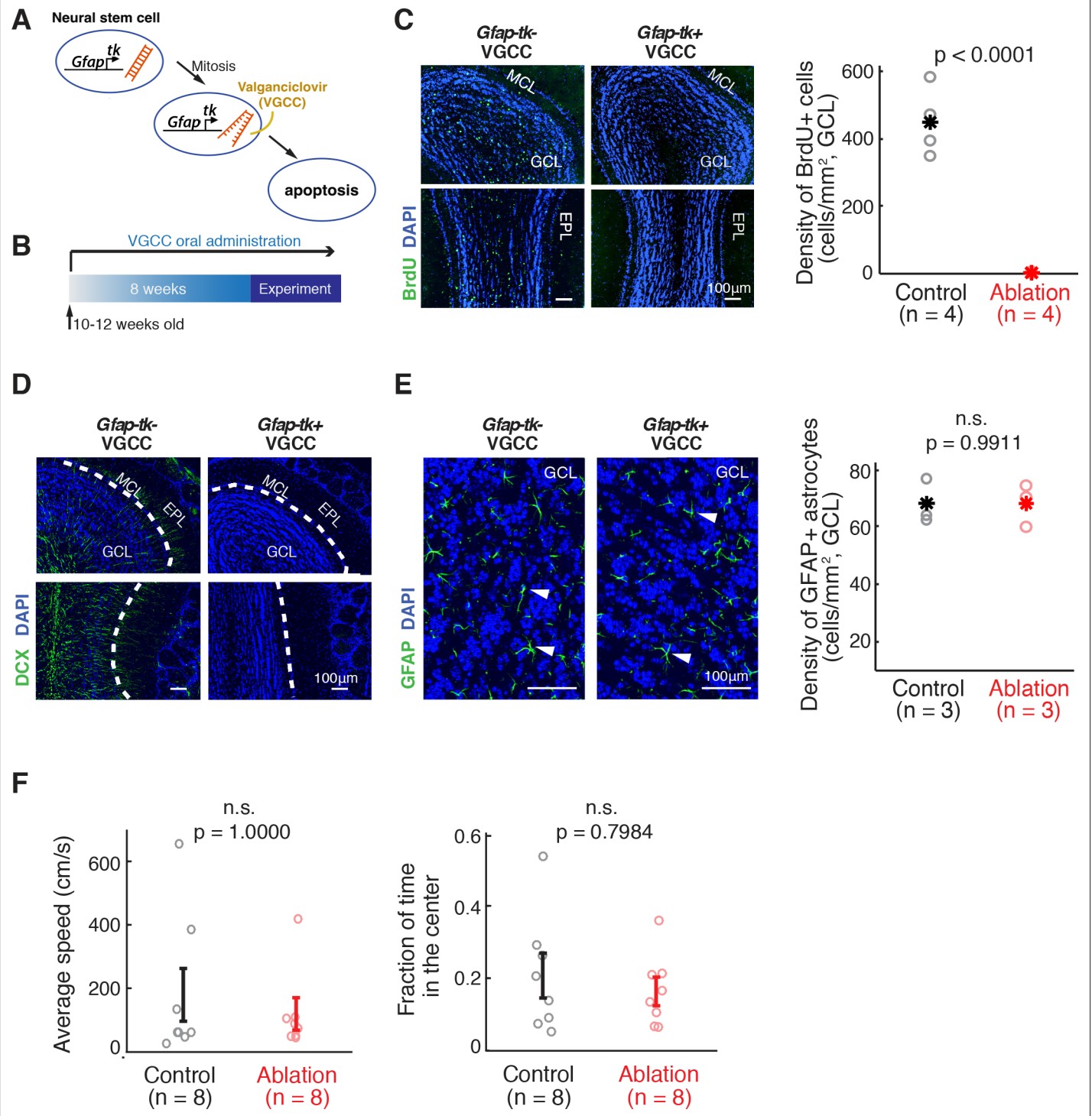

**Figure 1.** Inducible ablation of adult neurogenesis. (**A**) Pharmacogenetic ablation of ABNs. Valganciclovir (VGCC) induces apoptosis of *Gfap*-expressing mitotic neural stems cells, blocking the generation of ABNs. (**B**) Experimental timeline. Adult mice underwent 8 weeks of VGCC treatment before starting the behavioral task and imaging. (**C,D**) VGCC administration results in a near-complete ablation of ABNs in the olfactory bulbs of *Gfap-tk+* (ablation) mice. (**C**) BrdU labeling of olfactory bulbs of *Gfap-tk-* (control) and ablation mice administered with VGCC. Green: BrdU; Blue: DAPI; MCL: mitral cell layer; GCL: granule cell layer; EPL: external plexiform layer. Right: quantification of the BrdU-labeled cell density in control (n = 4, black) and ablation (n = 4, red) (p<0.0001, Wilcoxon rank sum test). Asterisks represent means. (**D**) Doublecortin (DCX) labeling of immature neurons in the olfactory bulbs showed similar results to BrdU labeling. Green: DCX; Blue: DAPI. (**E**) VGCC administration in ablation mice does not affect the density of GFAP astrocytes in the olfactory bulb. Left: GFAP immunostaining of astrocytes in the olfactory bulbs of control mice (left panel) and ablation mice

*Figure 1 continued on next page*

*Figure 1 continued*

(right panel) after 8 weeks of continuous treatment of VGCC. Arrows show examples of GFAP astrocyte cell bodies. Green: GFAP; Blue: DAPI. Right: No significant difference in the density of GFAP astrocytes in VGCC-administered control mice (n = 3, black) and ablation mice (n = 3, red) (mean ±S.E.M.; p=0.9911, Wilcoxon rank sum test). Asterisks represent means. (**F**) Open field test shows no deficiency in mobility in ablation mice (n = 8, red) compared to controls (n = 8, black). All error bars: mean ±S.E.M. Left: average speed (cm/s) (p=1.0000, Wilcoxon rank sum test). Right: fraction of time spent in the center area (p=0.7984, Wilcoxon rank sum test).

DOI: https://doi.org/10.7554/eLife.33006.002

The following figure supplement is available for figure 1:

**Figure supplement 1.** VGCC administration in ablation mice does not affect the density of GFAP astrocytes in the hippocampus.

DOI: https://doi.org/10.7554/eLife.33006.003

---

Given these results, we asked whether finer discrimination would reveal a deficit caused by ABN ablation. To address this question, we devised a difficult discrimination task in which 1 of 8 very similar mixtures, each with slightly varying ratios of ethyl-tiglate and heptanal, was presented in each trial. Four of the eight mixtures signaled left lick trials, while the other four mixtures signaled right lick trials (*Figure 2D*). After mice achieved expertise in the easy discrimination task, they were trained with this difficult discrimination task over 10 sessions. Although the performance of control animals was initially at chance level, it consistently improved over 10 sessions to achieve a success rate of 0.760 ± 0.017 in session 10. In contrast, ablation animals showed slower learning (comparison of linear regression slopes in individual animals, p=0.0100, Wilcoxon rank sum test), and their performance was significantly lower than that of the control animals (p(group)<0.0001, p(session)<0.0001, two-way ANOVA; *Figure 2E*).

The deficits observed in ablation animals were unlikely due to problems in motivation or licking ability, as both groups had the comparable fractions of answered trials (control vs. ablation, p(group) = 0.4101, two-way ANOVA; *Figure 2F*) and the comparable licking rates during reward consumption (control vs. ablation, p(group) = 0.1036, two-way ANOVA; *Figure 2G*). To confirm that mice were performing the task by using odorant stimuli as the cues and not other cues (such as potential differences in sounds of different odorant valves), we performed an additional session after the 10th session. In this test session, all odorants were replaced with the same 50H50E mixture while the contingency between odorant valves and correct lick side was maintained. The performance of both groups dropped to chance level in the test session (control: p=0.7241, ablation: p=0.6925, t test with chance level (0.5); *Figure 2E*, '50:50'), indicating that they were indeed relying on odorants as the cue. As an additional control, we trained a separate cohort of mice (control: n = 7; ablation: n = 6). After the pre-training with easy 2-odorant discrimination, these mice were trained with another version of 8-odorant discrimination task in which we used eight highly distinct mixture ratios ('easy 8-odorant discrimination'; *Figure 2—figure supplement 1A*). On the first day of training with the easy 8-odorant discrimination task, both control and ablation animals performed similarly, at the level equivalent to the easy 2-odorant discrimination (control: p=0.5350, ablation: p=0.2229, Wilcoxon rank sum test), which was significantly better than the first session of difficult 8-odorant discrimination (p(task)<0.0001, 3-way ANOVA; *Figure 2—figure supplement 1B*). Therefore, the deficits of ablation animals in the difficult 8-odorant discrimination were due to the requirement of fine discrimination and not due to the complexity of 8 different mixtures. In conclusion, mice without young ABNs are impaired in the difficult discrimination task requiring fine odorant discrimination.

## Suppression of hippocampal postnatally-born neurons alone or random ablation of GCs did not impair fine discrimination

The results above suggest that ABNs in the olfactory bulb are critical for fine odorant discrimination. However, the dentate gyrus (DG) of hippocampus is the other major niche for adult neurogenesis (*Gonçalves et al., 2016*; *Ming and Song, 2011*), and the *Gfap-tk* method suppresses adult neurogenesis in both the SVZ and DG. To address this issue, we adopted another transgenic method previously described that could specifically suppress postnatally-born DG neurons (*mGfap-Cre::Slc17a7-LoxP-STOP-LoxP-tetanus neurotoxin (LSL-TeNT)*, hereafter referred to as 'DG suppression', *Slc17a7* is also known as *vGlut1*, which is expressed in excitatory DG granule cells but not in inhibitory OB GCs) (*Sakamoto et al., 2014*). As ABNs are a subset of postnatally-born DG neurons, this DG suppression

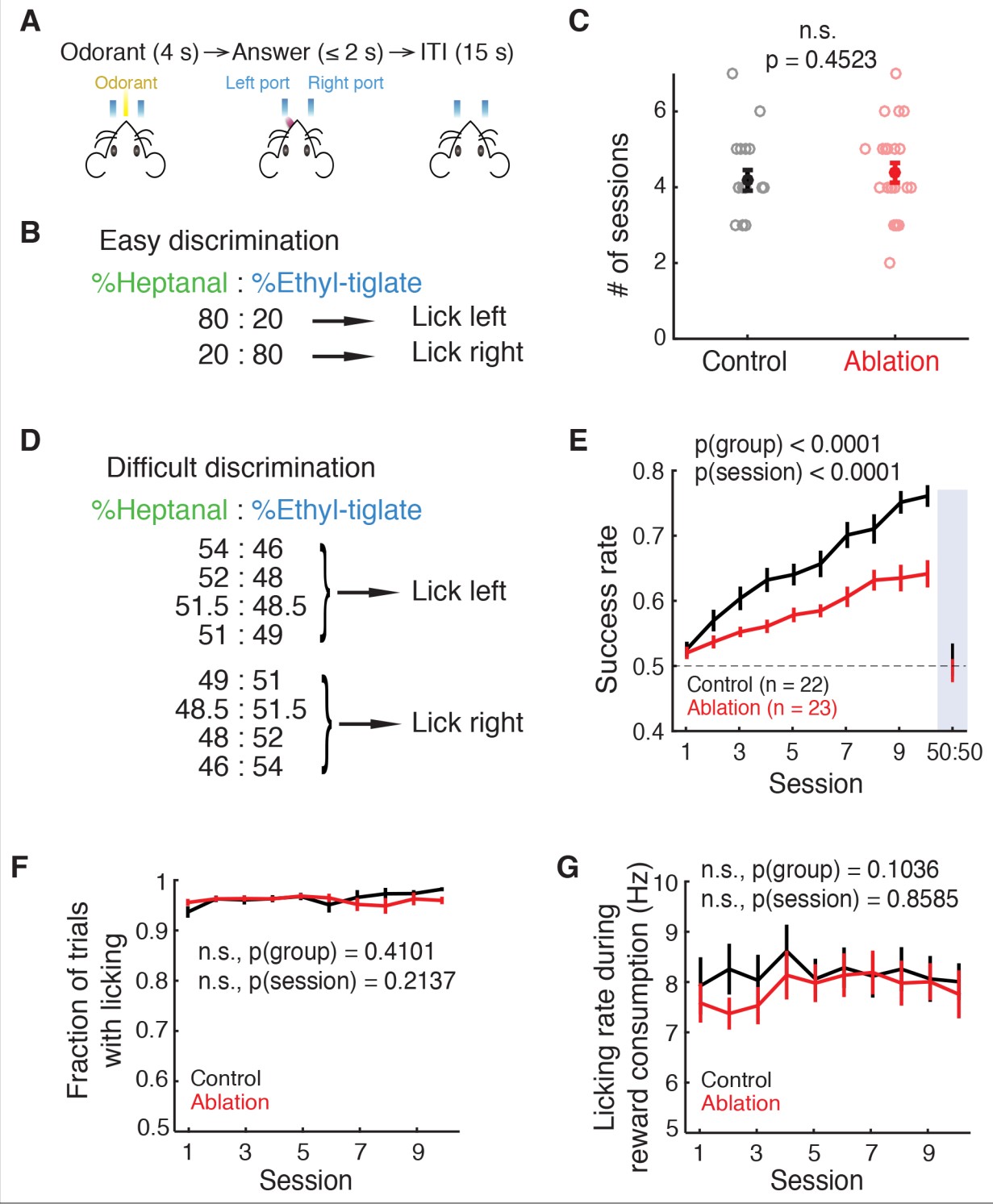

**Figure 2.** Ablation mice are impaired in difficult but not easy discrimination task. (**A**) Trial structure. (**B**) Easy discrimination task. Mice are trained to lick left in response to H80E20 (a mixture of 80% heptanal and 20% ethyl-tiglate) and lick right to H20E80. (**C**) Number of sessions required to reach expertise (>80% success rate) in the easy discrimination task. There is no significant difference between control (n = 22) and ablation (n = 23) groups (p=0.4523, Wilcoxon rank sum test). Learning curves are not shown since each animal was trained until expertise, resulting in varying durations of training. (**D**) Difficult discrimination task. One of the eight mixtures is presented pseudorandomly in each trial. (**E**) Fraction of correct trials over 10 sessions in the difficult discrimination task. Ablation animals exhibit impaired learning compared to control animals (p(group)<0.0001, p(session)<0.0001, two-way ANOVA). Both groups had equal and chance-level success rate in the test session with identical H50E50 mixtures, indicating that they were using odorants to guide their choice (described in Results). (**F**) Fraction of trials with answers (correct or incorrect) throughout

*Figure 2 continued on next page*

*Figure 2 continued*

the difficult task sessions. There is no difference between control and ablation (p=0.4101, two-way ANOVA). (**G**) Control and ablation had comparable lick rates during reward consumption (p=0.1036, two-way ANOVA). All error bars: mean ±S.E.M.

DOI: https://doi.org/10.7554/eLife.33006.004

The following figure supplement is available for figure 2:

**Figure supplement 1.** ABN ablation does not affect the performance in the easy 8-odorant discrimination.

DOI: https://doi.org/10.7554/eLife.33006.005

method targeted a much larger fraction (more than one third) of DG neurons (*Figure 3—figure supplement 1a and b*). Supporting the effective expression of TeNT in a large fraction of DG granule cells, immunostaining signal for VAMP2 in the mossy fiber terminals in CA3 was significantly reduced in the DG suppression mice (*Figure 3—figure supplement 1c*). These results are consistent with the idea that the 'DG suppression' strategy effectively suppressed most, if not all, of postnatally-born DG granule cells including DG ABNs, even though it is difficult to directly demonstrate the inhibition of synaptic release from all DG ABNs.

During the same easy and difficult discrimination tasks described in *Figure 2*, the DG suppression mice showed comparable performance to the control group in both easy and difficult discrimination tasks (control vs. DG suppression; easy discrimination: number of sessions to reach expertise, p=0.8407, Wilcoxon rank sum test; difficult discrimination: p(group)=0.2663, p(session)<0.0001, 2-way ANOVA; mean ±S.E.M.; *Figure 3A,B*). These results indicate that DG postnatally-born neurons that include ABNs are not essential for fine olfactory discrimination and the behavioral impairment in *Gfap-tk+* ablation group was primarily caused by the absence of ABNs in the olfactory bulb. We do note, however, that a formal possibility remains that a small fraction of DG ABNs that may have been spared in the 'DG suppression' method may partially contribute to the behavioral deficits of the *Gfap-tk+* ablation group.

Next we considered two alternative possibilities underlying the behavioral deficit in the ablation animals. First, it is possible that ablation of any inhibitory neurons in the olfactory bulb may lead to similar deficits. Second, olfactory fine discrimination may be particularly sensitive to ABN ablation. To distinguish these possibilities, we sought to ablate a random subset of GCs regardless of their age. To this goal, we first estimated the degree of neuron loss in the ABN ablation animals by quantifying cell density in the granule cell layer (GCL) using DAPI labeling. This indicated a 10.02% reduction of total cell numbers through ABN ablation by the end of the behavioral tasks (control: n = 3, ablation: n = 4; *Figure 3C,D*). To achieve a similar level of neuron ablation without specifically targeting ABNs ('random ablation'), we bilaterally injected a mixture of diluted AAV2/1-CMV-Cre and AAV2/1-EF1a-FLEX-taCaspase3 (*Yang et al., 2013*) into the center of the olfactory bulb. Post hoc DAPI staining and quantification ~1 month after injections revealed a 19.66% reduction in GCL cell density compared with un-injected animals (injected (n = 11) vs. un-injected control (n = 7): p<0.0001, Wilcoxon rank sum test; *Figure 3E,F*), with no significant change in the width of GCL (injected vs. un-injected control: p=0.9499, Wilcoxon rank sum test; *Figure 3G*). Thus, our random ablation eliminated a larger number of cells than our ABN ablation. Importantly, we found no change in Doublecortin immunostaining, indicating that our random ablation did not affect subsequent adult neurogenesis (injected (n = 7) vs. uninjected (n = 7) control, p=0.9015, Wilcoxon rank sum test; *Figure 3H,I*). We trained these random ablation animals starting at 1 month after the injections. Random ablation animals exhibited normal performance in the easy discrimination task (control vs. random ablation, number of sessions to reach expertise, p=0.4430, Wilcoxon rank sum test; mean ±S.E.M., *Figure 3J*). In the difficult discrimination task, the random ablation group initially showed slower learning than the control group, but they eventually reached the performance level that was statistically indistinguishable from the controls and significantly better than ABN ablation animals (Sessions 1–10: random ablation (n = 11) vs. control (n = 22), p=0.0034, random ablation vs. ABN ablation, p<0.0001; Sessions 1–5: random ablation vs. control, p=0.0039, random ablation vs. ABN ablation, p=0.4405; Sessions 6–10: random ablation vs. control, p=0.2028, random ablation vs. ABN ablation, p<0.0001; 2-way ANOVA; mean ±S.E.M.; *Figure 3K*). Together with the observation that the random ablation eliminated more cells than in ABN ablation, these results support the notion that young ABNs have a privileged role in mediating fine olfactory discrimination.

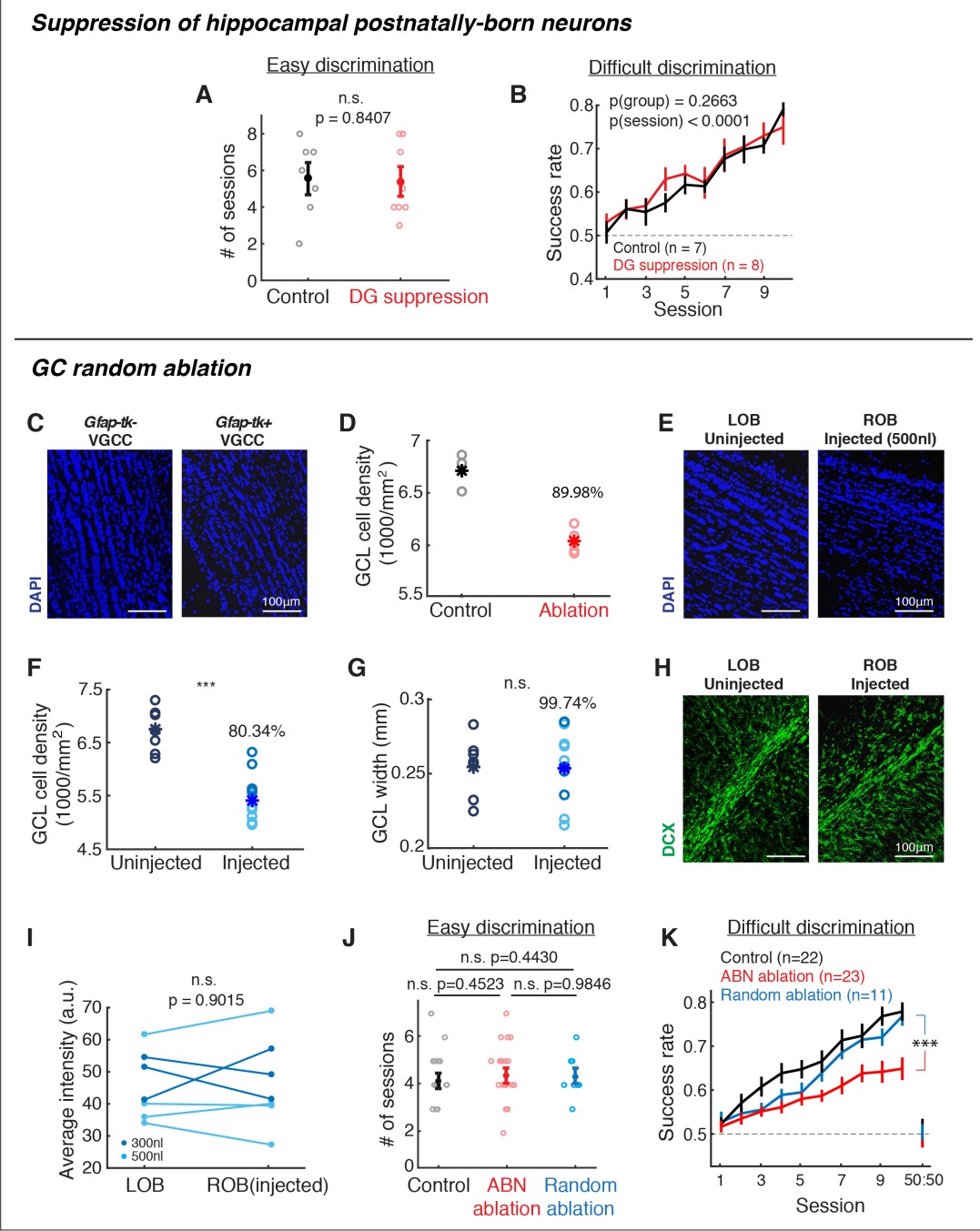

**Figure 3.** Suppression of hippocampal postnatally-born neurons or random ablation of GCL neurons did not cause the same behavioral deficit as *Gfap-tk* mice. (**A,B**) Behavioral performance of control (black) vs. hippocampal DG postnatally-born neuron suppression (red) groups. (**A**) Number of sessions required to reach expertise (>80% success rate) for the easy discrimination task. There is no significant difference between control (n = 7) and DG suppression (n = 8) groups (p=0.8047, Wilcoxon rank sum test). (**B**) Fraction of correct trials over 10 sessions in the difficult discrimination task. There is no significant difference between control (n = 7) and DG suppression groups (n = 8; p(group)=0.2663, two-way ANOVA). (**C,D**) Quantification of GCL neuron reduction after VGCC treatment. (**C**) Post hoc DAPI (blue) labeling shows that after ~2 months of VGCC treatment, *Gfap-tk*+ (right) ablation group have a lower cell density in the GCL compared to *Gfap-tk*- (left) controls. (**D**) GCL cell density in control (n = 3, black) and ablation (n = 4, red). Asterisks represent means. On average, density (ablation)/density (control) = 89.98%, indicating that there was a 10.02% reduction in cell density. (**E,F,G**) Random cell ablation in GCL by injecting a combination of AAV2/1-CMV-Cre and AAV2/1-EF1a-FLEX-taCasp viruses caused a reduction in cell density, without changing the GCL size or affecting olfactory adult neurogenesis. (**E**)

*Figure 3 continued on next page*

*Figure 3 continued*

DAPI labeling 10 days after right OB unilateral injection shows a reduced cell density in GCL of the injected right OB (right) compared to the uninjected left OB (left). Blue: DAPI. (**F**) Post hoc quantification of GCL DAPI signal density 1.5 months after 300 nl (n = 4, dark blue) or 500 nl (n = 7, light blue) viral cocktail bilateral injection compared to uninjected control (n = 7) (injected vs. unjected control: p<0.0001; Wilcoxon rank sum test). Asterisks represent means. (**G**) There is no difference in GCL width with or without viral ablation (injected vs. uninjected control: p=0.9499; Wilcoxon rank sum test). Asterisks represent means.. (**H,I**) Random ablation method did not affect olfactory adult neurogenesis. (**H**) Doublecortin (DCX) labeling of immature neurons in uninjected (left) or injected (right) OBs 10 days after right OB unilateral injection. Green: DCX. (**I**) Average DCX signal intensity of RMS-periRMS area in the OB. There is no difference between un-injected and injected OBs (p=0.9015, Wilcoxon rank sum test), indicating that the injections did not affect subsequent adult neurogenesis. Dark blue: 300 nl injection; light blue: 500 nl injection. (**J,K**) Behavioral performance of random GCL cell ablation group (blue, n = 11), compared to control (black) and ABN ablation (red) groups shown in *Figure 2C,E*. (**J**) Number of sessions required to reach expertise (>80% success rate) for the easy discrimination task. There is no significant difference between random ablation group and control or ABN ablation groups (random ablation vs. control: p=0.4430; random ablation vs. ABN ablation: p=0.9846; Wilcoxon rank sum test). (**K**) Fraction of correct trials in the difficult discrimination task. For all 10 sessions, random ablation group is significantly different from other two groups (random ablation vs. control, p<0.0001; random ablation vs. ABN ablation, p<0.0001). For sessions 1–5, random ablation is worse than control (p=0.0124), but not different from ABN ablation (p=0.2528); for sessions 6–10, random ablation is better than ABN ablation (p<0.0001), but not different from control (p=0.2575). Mean ±S.E.M., two-way ANOVA.

DOI: https://doi.org/10.7554/eLife.33006.006

The following figure supplement is available for figure 3:

**Figure supplement 1.** Suppression of hippocampal postnatally-born neurons with *mGfap-Cre*::*vGlut1-LSL-TeNT* targeted a larger fraction of DG neurons than ABN ablation.

DOI: https://doi.org/10.7554/eLife.33006.007

## ABN ablation affects MC population coding during difficult discrimination

To investigate the neural basis of the impaired discrimination in ABN ablation animals, we monitored the activity of MCs in ablation and control animals using two-photon calcium imaging. We utilized the transgenic mouse line *Cdhr1-Cre* (*Cdhr1* is also known as *Pcdh21*), which expresses *Cre* specifically in the olfactory bulb principal neurons. We injected AAV1-hsyn-FLEX-GCaMP6f in the right olfactory bulb of *Gfap-tk$^{+/-}$*::*Cdhr1-Cre* (ablation) or *Gfap-tk$^{-/-}$*::*Cdhr1-Cre* (control, littermates) animals to specifically express GCaMP6f in mitral/tufted cells (*Figure 4A*). After training with the easy discrimination task, these mice were trained with the difficult discrimination task while we imaged the ensemble activity of MCs (control: n = 12, ablation: n = 10; *Figure 4B,C*).

Individual MCs showed odorant-specific responses with an increase or decrease in GCaMP6f fluorescence (*Figure 4D*). To quantify the discriminability of the eight mixtures by the MC ensembles, we performed decoder analysis (*Chu et al., 2016*) which attempts to decode the odorant on each trial based on the population activity of individual MCs during the odorant period (Materials and methods). If the decoded odorant matched the actual odorant, the trial was scored as correct. We found that decoder accuracy was significantly better than chance (0.125) in both control and ablation groups (control: p<0.001, ablation: p<0.001, Student's t-test; *Figure 4E*). However, the decoder accuracy was higher in control animals than in ablation animals (control vs. ablation, p(group)<0.05, two-way ANOVA; *Figure 4E*). These results indicate that MC responses to different mixtures are more ambiguous in ablation animals than in control. Next we asked whether the separation of mitral cell odor representations is sensitive to the similarity of odor mixtures. To address this, we performed a pairwise decoder analysis in which we built a decoder to decode the mixture identity for each pair of the eight mixtures. Here we defined the 'contrast' between each pair of mixtures as the difference in the percentage of heptanal (*Figure 4F*). For example, the contrast between 52H48E and 48.5H51.5E is 3.5 (=52–48.5). We found that within each of the control and ablation groups, there was a positive correlation between pairwise decoder accuracy and the contrast between the mixtures (control: r = 0.1862, p<0.01, ablation: r = 0.3029, p<0.0001, Pearson correlation; *Figure 4G*). Although the decoder performance of the control group was generally better than the ablation group, the difference was

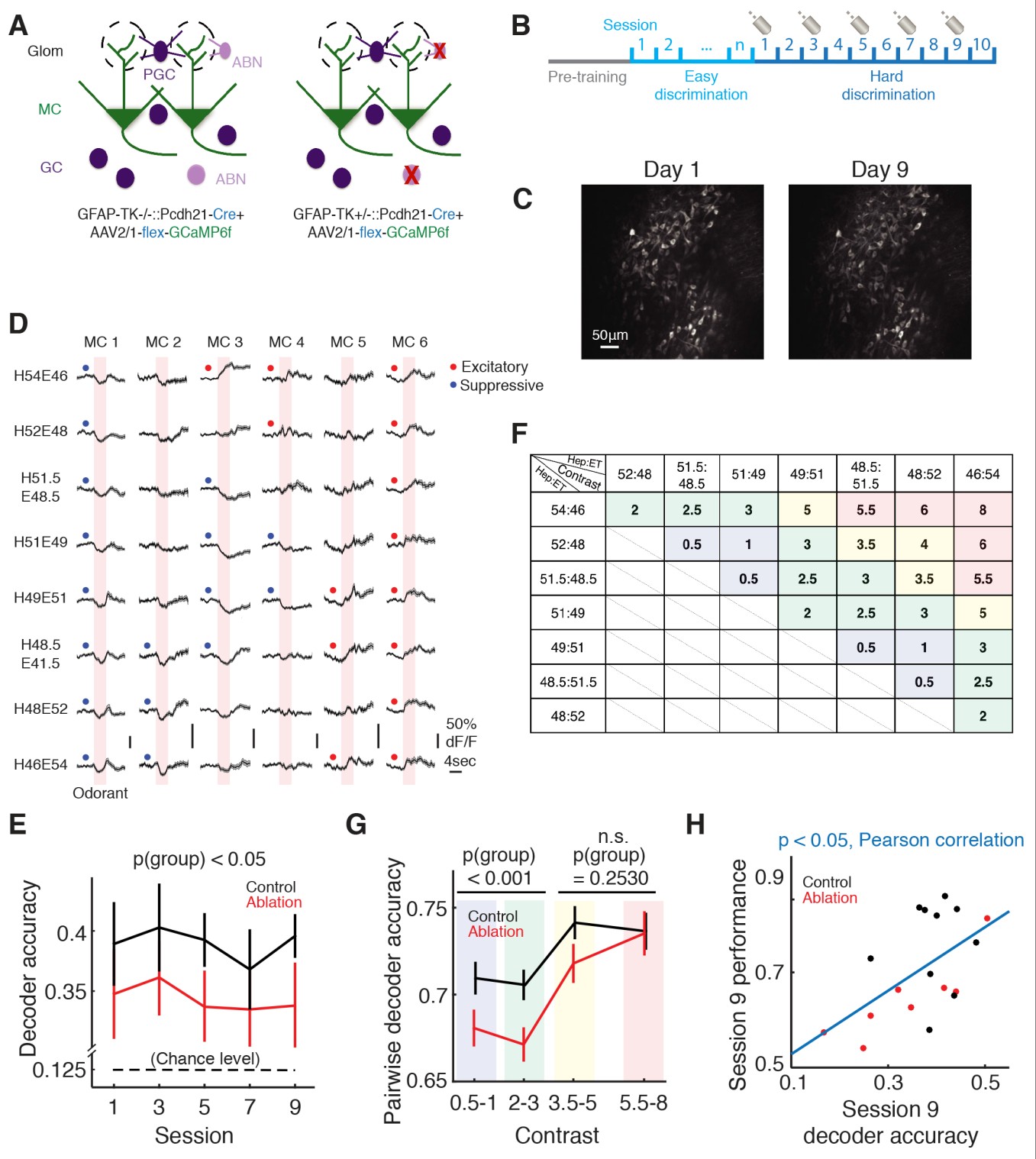

**Figure 4.** Imaging of mitral cell responses during the difficult discrimination task. (**A**) Schematic of the olfactory bulb. AAV2/1-flex-GCaMP6f was injected into the olfactory bulb to express GCaMP6f in mitral/tufted cells in both control (*Gfap-tk*-/-::*Cdhr1-Cre*) and ablation (*Gfap-tk*+/-::*Cdhr1-Cre*) animals. VGCC was administered continuously to both groups, resulting in ABN ablation in the ablation group (right) but not control group (left). (**B**) Imaging timeline. After the pretraining period, mice were trained with the easy discrimination task until they reached expertise (>80% fraction correct within a session). The mice were then trained to perform the 8-odorant difficult discrimination task for 10 sessions with two-photon imaging on sessions 1, 3, 5, 7 and 9. (**C**) A field of the same mitral cell (MC) population on first day of imaging (left) and 8 days later (right). (**D**) Odorant responses (mean ±S.E.M.) of six example MCs during the first day of the difficult discrimination task. Pink areas denote the 4 s odorant period. Red and blue

*Figure 4 continued on next page*

*Figure 4 continued*

dots indicate significant excitatory and suppressive responses respectively. (**E**) Population decoder accuracy during the difficult discrimination task in ablation animals (n = 10) is significantly worse than control animals (n = 12) (mean ±S.E.M., p<0.05, two-way ANOVA). Black broken line indicates the chance level (0.125). (**F**) Table of contrast values between odorant pairs for the eight odorants used in the difficult discrimination task. Different color shades in the boxes match with the binning of the contrast values in **G**. (**G**) Pairwise decoder accuracy during the difficult discrimination task plotted as a function of binned odorant pair contrasts (mean ±S.E.M.). Control is significantly better than ablation for smaller contrasts (≤3, p<0.001, two-way ANOVA) but not for larger contrasts (>3, p=0.2530, two-way ANOVA). (**H**) Behavioral performance in session 9 correlates with decoder accuracy of the session (mean ±S.E.M., p<0.05, Pearson correlation).

DOI: https://doi.org/10.7554/eLife.33006.008

The following figure supplement is available for figure 4:

**Figure supplement 1.** Categorical associations with choices do not influence the pattern separation of MC population responses.

DOI: https://doi.org/10.7554/eLife.33006.009

more prominent in mixture pairs with smaller contrasts (≤3) (control vs. ablation, pairs with contrasts ≤3: p(group)<0.001, pairs with contrasts >3: p(group)=0.2530, two-way ANOVA; *Figure 4G*). These results suggest that ABN ablation causes the separation of MC population responses to similar odorants to be less robust, possibly underlying the behavioral deficits in fine discrimination. Consistent with this notion, the decoder accuracy of individual animals positively correlated with their behavioral performance (p<0.05, Pearson correlation; *Figure 4H*).

We next asked whether the categorical associations with the left or right lick side influenced the MC population responses. To investigate this, we analyzed the decoder performance based on the difference ('contrast') between the two odorants and whether the two odorants were associated with the same or different lick sides (*Figure 4—figure supplement 1*). The decoder performance was better when the two odorants were more distinct. However, the decoder performance was not affected by whether the two odorants were associated with the same or different lick sides (control: p=0.4727, ablation: p=1.0000, same vs. different associations in the bin 'contrast = 2–3', Wilcoxon rank sum test; *Figure 4—figure supplement 1*). Thus, in our dataset, we found no evidence that MC responses categorized odorants based on the associated choices.

## ABNs are essential for suppressive responses of mitral cells

To investigate the basis for the decreased decoder accuracy in ABN ablation animals, we analyzed the responses of individual mitral cells to the eight mixtures. We quantified two measures; the first is the fraction of MCs that responded to at least one mixture, and the second is the fraction of responsive MC-odorant pairs out of all MC-odorant pairs. We found that the fraction of MCs responsive to at least one mixture and the fraction of responsive MC-odorant pairs were both consistently lower in ablation animals compared to control (control vs. ablation, fraction of cells: p<0.001, fraction of cell-odorant pairs: p<0.01, two-way ANOVA; *Figure 5A*). As a MC can respond to an odorant with increased or decreased activity, we next quantified excitatory and suppressive responses separately. This analysis showed that the excitatory response fraction was not significantly affected by ABN ablation (control vs. ablation, fraction of cells: p=0.0813, fraction of cell-odorant pairs: p=0.6039, two-way ANOVA; *Figure 5B*). Instead, the decreased responses in ablation animals were primarily due to decreases in suppressive responses (control vs. ablation, fraction of cells: p<0.0001, fraction of cell-odorant pairs: p<0.0001, two-way ANOVA; *Figure 5C*), suggesting that the net effect of ABNs on MC ensembles is inhibitory.

The decrease in suppressive but not excitatory responses of MCs in ablation animals raises the possibility that the decreased suppressive responses may underlie the reduced decoder accuracy in ablation animals. Therefore we explored the relationships of excitatory and suppressive MC responses with decoder accuracy and behavior. We found that the fraction of total (excitatory and suppressive) responses positively correlates with decoder accuracy (fraction of cells: p<0.01, fraction of cell-odorant pairs: p<0.01, Pearson correlation) and behavioral performance (cells: p<0.01; cell-odorant pairs: p<0.05; *Figure 6A,B*). When we only included excitatory responses, however, this relationship was not significant (decoder accuracy, cells: p=0.2089, cell-odorant pairs: p=0.1272; performance, cells: p<0.05, cell-odorant pairs: p=0.1459; *Figure 6C* left and *Figure 6D* left). Instead, the fraction of suppressive responses significantly correlated with both decoder accuracy and behavioral

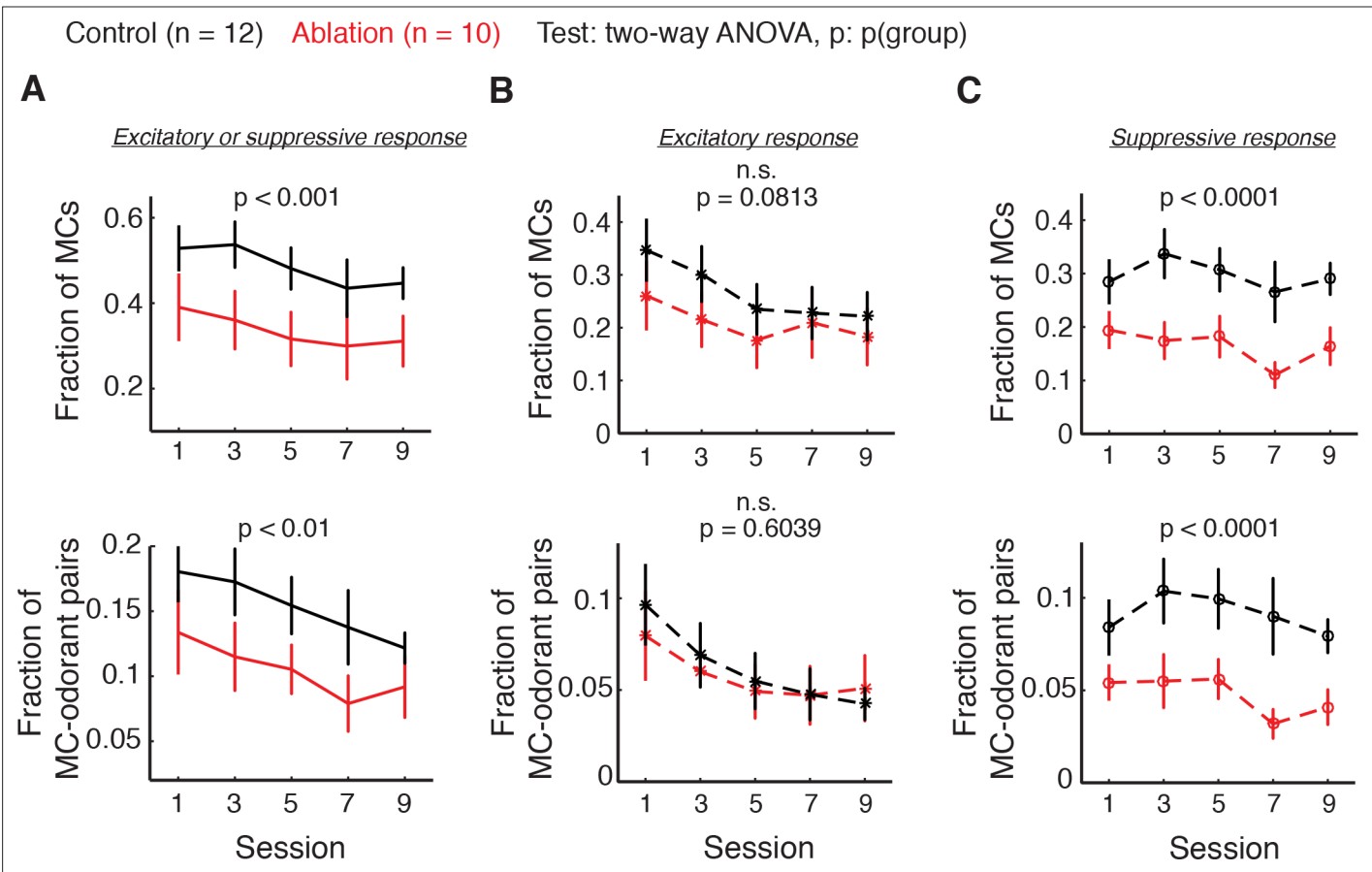

**Figure 5.** Fraction of responsive MCs (top) and responsive MC-odorant pairs (bottom) during the difficult discrimination task. Control: n = 12; ablation: n = 10. Mean ±S.E.M. (**A**) MCs with either excitatory or suppressive responses. Fraction of MCs: p<0.001; fraction of MC-odorant pairs: p<0.01. (**B**) MCs with excitatory responses. Fraction of MCs: p=0.0813; fraction of MC-odorant pairs: p=0.6039. (**C**) MCs with suppressive responses. Fraction of MCs: p<0.0001; fraction of MC-odorant pairs: p<0.0001. All p values are for two-way ANOVA, control vs. ablation. Note that a given MC can have excitatory responses to some odorants and suppressive responses to others, and thus the total response is not necessarily a sum of excitatory and suppressive.

DOI: https://doi.org/10.7554/eLife.33006.010

performance (decoder accuracy, cells: p<0.01, cell-odorant pairs: p<0.01; performance, cells: p<0.05, cell-odorant pairs: p<0.05; **Figure 6C** right and **Figure 6D** right). Together these results suggest that young ABNs are essential for high levels of suppressive responses of MCs, which significantly contribute to odorant discriminability by MC population responses.

## The necessity of ABNs depends on task engagement

Inhibitory neurons in the olfactory bulb, including ABNs, are major targets of extensive glutamatergic and neuromodulatory projections from higher brain areas. These centrifugal projections are suggested to be sensitive to brain states (*Gilbert and Sigman, 2007*; *Rothermel and Wachowiak, 2014*). Therefore it is tempting to hypothesize that the impact of ABN functions is sensitive to behavioral states such as task engagement. We reasoned that, if this is the case, the differences in the MC responses of control and ablation animals described above would be less pronounced when the mice were not engaged in the task.

To test this idea, we performed a new experiment in which another cohort of mice were passively exposed to odorants (control passive: n = 10; ablation passive: n = 7). Except for the lack of task engagement, all the other conditions were kept identical to the task condition, including VGCC treatment, water restriction, odorant stimulation protocol, and odorant identity (4 sessions of 2 'easy discrimination' odorants followed by 10 sessions of 8 'difficult discrimination' odorants). Imaging was

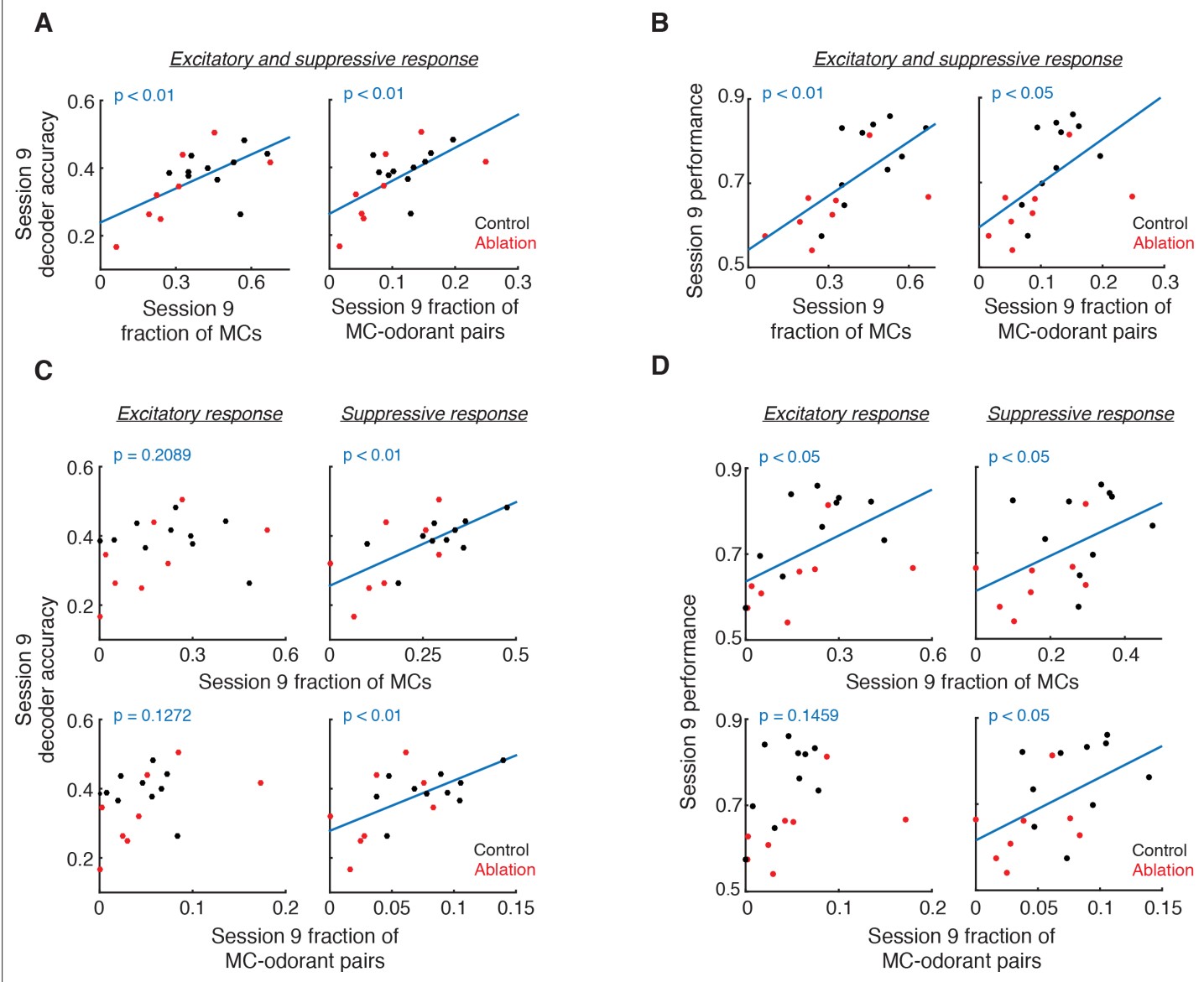

**Figure 6.** Relationship of responsive MC fractions with MC ensemble discriminability and behavioral performance. (**A**) Session 9 decoder accuracy correlates with fraction of responsive MCs (left, p<0.01, Pearson correlation) and MC-odorant pairs (right, p<0.01, Pearson correlation). (**B**) Session 9 behavioral performance correlates with fraction of responsive MCs (left, p<0.01, Pearson correlation) and MC-odorant pairs (right, p<0.05, Pearson correlation). (**C**) Session 9 decoder accuracy correlates with suppressive but not excitatory MC responses. Top left: excitatory MCs, p=0.2089; bottom left: excitatory MC-odorant pairs, p=0.1272; top right: suppressive MCs, p<0.01; bottom right: suppressive MC-odorant pairs, p<0.01, Pearson correlation. (**D**) Session 9 behavioral performance correlates with MC suppressive responses. Top left: excitatory MCs, p<0.05; bottom left: excitatory MC-odorant pairs, p=0.1459; top right: suppressive MCs, p<0.05; bottom right: suppressive MC-odorant pairs, p<0.05, Pearson correlation.

DOI: https://doi.org/10.7554/eLife.33006.011

performed during the passive experience of the eight difficult discrimination odorants that are identical to the task condition. Strikingly, in this passive condition, the fractions of MCs showing excitatory and suppressive responses were no longer statistically distinguishable between ablation and control animals (control passive vs. ablation passive, fraction of cells: total, p=0.6162; excitatory, p=0.7625, suppressive, p=0.2620; fraction of cell-odorant pairs: total, p=0.2043; excitatory, p=0.8691, suppressive, p=0.0625, two-way ANOVA; *Figure 7A–C*). These results suggest that MC responses, mainly suppressive responses, are increased in a task engagement-dependent manner. We further tested this notion with a linear regression model (Materials and methods). The interaction term for the genotype (control vs. ablation) and the condition (task vs. passive) was statistically significant for suppressive

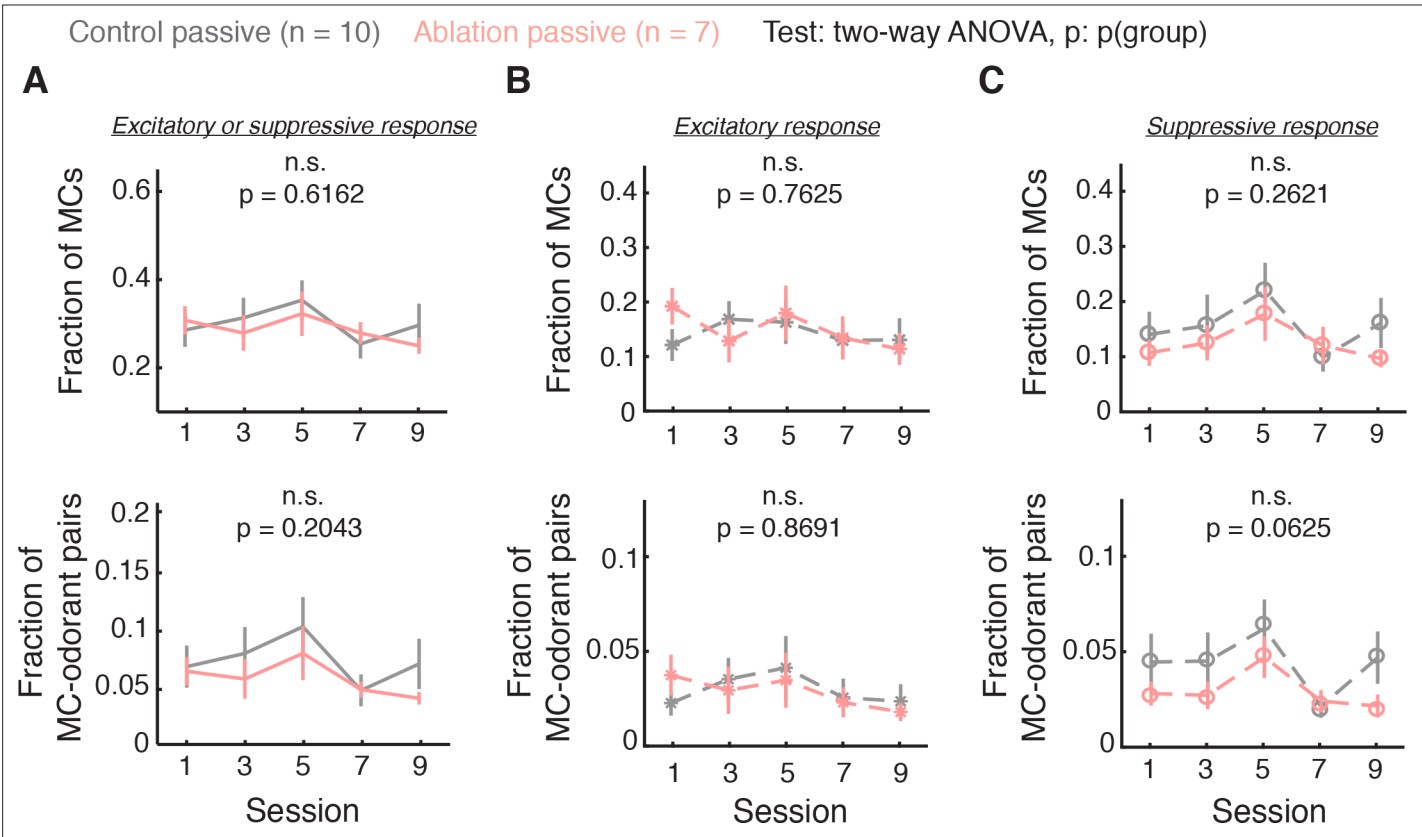

**Figure 7.** Task engagement enhances MC suppressive responses through ABNs. (**A–C**) Fraction of responsive MCs (top) and responsive MC-odorant pairs (bottom) during passive exposure. Control passive: n = 10; ablation passive: n = 7. Mean ±S.E.M. (**A**) MCs with either excitatory or suppressive responses. Fraction of MCs: p=0.6162; fraction of MC-odorant pairs: p=0.2043. (**B**) MCs with excitatory responses. Fraction of MCs: p=0.7625; fraction of MC-odorant pairs: p=0.8691. (**C**) MCs with suppressive responses. Fraction of MCs: p=0.2620; fraction of MC-odorant pairs: p=0.0625. All p values are for two-way ANOVA, control passive vs. ablation passive.

DOI: https://doi.org/10.7554/eLife.33006.012

The following figure supplement is available for figure 7:

**Figure supplement 1.** Fractions of responsive MCs during difficult discrimination task and passive exposure.

DOI: https://doi.org/10.7554/eLife.33006.013

(fraction of cells: p=0.0101; fraction of cell-odorant pairs: p=0.0239), but not excitatory (fraction of cells: p=0.1162; fraction of cell-odorant pairs: p=0.7509) responses. This result indicates that the effect of task engagement on suppressive, but not excitatory, responses is significantly larger in control animals than in ablation animals, supporting a role for young ABNs in enhancing suppressive responses in a task engagement-dependent manner.

Based on these observations, we propose that behavioral states strongly modulate MC activity and in particular suppressive responses, facilitating olfactory discrimination during task engagement. Importantly, this state-dependent enhancement of suppressive responses during task engagement requires young ABNs.

## Discussion

In this study, we ablated young ABNs in adult mice and investigated the consequences on their behavior and MC activity to probe the functional significance of adult neurogenesis. To our knowledge, this is the first study to record MC activity in ABN ablation animals during a behavioral task that reveals their impaired discrimination ability. The results provide a glimpse of the specific functions of ABNs.

## Behavioral consequences of adult neurogenesis ablation

Previous studies on the effect of ABN ablation on olfactory behavior have reported inconsistent results. These discrepancies may be due to differences in ablation methods as well as task demands. In this study, we adopted an inducible, genetic ablation method and we confirmed that this method almost completely eliminated ABNs. We and others also have found no evidence of non-specific effects on other cell types (*Cummings et al., 2014*; *Snyder et al., 2011*). Thus, this method can ablate young ABNs with high specificity and efficiency, allowing us to investigate its consequences on olfactory behavior and odor representations.

We established an olfactory discrimination two alternative choice task with two levels of difficulty. The operant and symmetric nature of the task allowed us to focus on the discrimination ability of individual animals, as opposed to spontaneous discrimination or asymmetric go/no-go tasks in which motivational states are difficult to control. Ablation mice were perfectly capable of discriminating conspicuously different odorants in this task, indicating that young ABNs are not necessary for basic odor processing, consistent with many previous studies. For difficult discrimination, we used eight binary mixtures of similar ratios applied pseudorandomly in each trial. Previous studies showed that tasks involving multiple similar odorants (*Rinberg et al., 2006*; *Uchida and Mainen, 2003*) delivered randomly (*Zariwala et al., 2013*) are more difficult than two-odorant tasks. This difficult condition revealed a robust impairment of ABN ablation animals in odor discrimination.

Importantly, our ablation method affects adult neurogenesis in both SVZ and hippocampus DG. However, DG-only suppression of ABNs did not produce the deficits in olfactory discrimination. Therefore, we conclude that young ABNs in the olfactory bulb are essential for fine discrimination of odorants.

Ablation of any neurons in the olfactory bulb may be expected to lead to deficits in olfactory behaviors. We find that ABNs less than 10–12 weeks old constitute ~10% of all GCs, consistent with a previous report (*Imayoshi et al., 2008*). When we ablated a larger fraction (~20%) of GCs randomly without regard to their age, majority of which were presumably mature GCs, the behavioral impairment was much more subtle. These results support the notion that young ABNs have a unique role in fine olfactory discrimination. However, we acknowledge an important caveat of the random ablation experiments in that the ablation was probably concentrated closer to the injection site and largely spared periglomerular cells, while the *Gfap-tk* method targeted ABNs in both GC and glomerular layers.

## Incorporation of inhibitory ABNs facilitate olfactory bulb pattern separation

To explore the potential neural basis underlying the behavioral impairment of ABN ablation animals, we used two-photon imaging to record the activity of populations of MCs (*Kato et al., 2012*) during the task performance, and analyzed data within the entire 4 s stimulus period. We are aware of the previous reports demonstrating the importance of finer time-scale dynamics of MC responses (*Abraham et al., 2004*; *Resulaj and Rinberg, 2015*; *Rinberg et al., 2006*; *Uchida and Mainen, 2003*; *Wachowiak, 2011*), which is not accessible with the temporal resolution of our approach. In fact, in certain reaction time tasks, the responses within the first 100 ms of odorant onset are sufficient for discrimination. However, we argue that it is unlikely that the responses within the first 100 ms of odorant onset explain the entirety of odor representations important for odor perception. This may especially be the case in conditions such as the task used here in which mice are not encouraged to react as quickly as possible. Furthermore, our approach affords a unique opportunity to record the activity of a few dozens of MCs longitudinally, allowing us to assess MC ensemble coding in ablation and control animals.

We performed MC calcium imaging during the difficult discrimination task to explore the potential neural basis underlying the behavioral impairment in ablation animals. This experiment revealed that the behavioral impairment in ablation animals accompanied a reduced separation of representations of similar odorants by MC ensembles as shown by the decoder analysis. Moreover, we found that the reduced separation of odor representations in ablation animals involved a preferential reduction of MC suppressive responses. The degree of reduction is related to the decoder performance, as the fraction of MC suppressive responses significantly correlated with the decoder accuracy and

behavioral performance, supporting the importance of MC suppressive responses. Considering that ABNs exert inhibitory modulation onto MCs, these results together suggest that the ablation of ABNs caused a reduction in MC suppressive responses, which in turn affected the discriminability of MC ensembles. In normal animals, excitatory inputs from sensory neurons combined with local inhibitory control would allow MCs to respond to odorants in both excitatory and suppressive manners (*Yokoi et al., 1995*). The bidirectionality of responses effectively increases the dynamic range of MC responses and would contribute to an enhanced separation of representations of similar odors.

Consistently, a modeling study simulated the effect of adult neurogenesis on the olfactory bulb circuitry with excitatory sensory inputs, local GC inhibition and MC outputs and predicted that a constant arrival and activity-dependent survival of ABNs are sufficient to separate MC representations of very similar odorants in an experience-dependent manner (*Cecchi et al., 2001*). A continuous recruitment of new ABNs allows the bulb to adapt to changes in the olfactory environment. Our current results indicate that such a mechanism is particularly sensitive to behaviorally significant experience such as engagement in difficult discrimination.

A prediction of our model is that MC suppressive responses and discriminability would be less affected (more similar to control animals) in the random ablation animals than in the ABN ablation animals. We made a great effort to test this possibility, but our effort to image MC activity in random ablation animals has so far been unsuccessful. Therefore this remains a question to be addressed in the future.

We note that the decoder performance was relatively stable throughout imaging, in contrast to our recent report in which the decoder performance improved during difficult discrimination learning and correlated with behavioral choice on a trial-by-trial basis (*Chu et al., 2016*). The apparent discrepancy likely stems from the fact that Chu et al. investigated changes of representations of novel odorants over time, while in the current study, mice had already been familiarized with the odorants, albeit at different mixture ratios, during the easy discrimination task prior to imaging. Despite the stable decoder accuracy in our current study, the degree of separation of mitral cell responses predicted the final behavioral performance, which is consistent with a previous report (*Gschwend et al., 2015*). It is also noteworthy that we did not detect an effect of categorical associations with lick sides on the degree of pattern separation (*Figure 4—figure supplement 1*). One interpretation of these results is that the olfactory bulb performs pattern separation based on the statistics of the olfactory environment (*Chu et al., 2016*) and not associations. The downstream areas (e.g. cortex) may be able to perform sensory-motor associations more efficiently when olfactory bulb outputs are more decorrelated. Another implication of these results is that there are likely multiple circuit origins of behavioral errors. When MC discriminability is a limiting factor for behavior, MC decoder accuracy may correlate with behavioral choice on each trial. However, in the current task, the downstream areas (e.g. olfactory cortex) and their ability to associate MC outputs with behavioral choice may be a main source of behavioral variability.

## Olfactory bulb as a state-dependent filter of odorant information

We found that the abundance of MC suppressive responses is highly sensitive to behavioral states. MC suppressive responses are increased during wakefulness compared to anesthetized states, and task engagement further enhances suppressive responses. This is consistent with a previous report stating that suppressive responses in MCs became more prominent during task engagement as opposed to passive exposure (*Fuentes et al., 2008*). Our results further extend these findings and demonstrate that the task engagement-dependent enhancement of suppressive responses is facilitated by young ABNs. During task engagement, ABN ablation animals have fewer suppressive responses than control animals. This difference in suppressive responses was absent during passive exposure. The sensitivity to task engagement may explain the findings from a previous study that broad GC inactivation has only mild effects on MC responses under anesthesia and passive wakefulness (*Fukunaga et al., 2014*). It is known that ABNs receive centrifugal synaptic and neuromodulatory inputs from multiple brain areas, and these inputs can vary depending on brain states. Thus, the state-dependence of the functional role of ABNs can be better appreciated considering previously reported phenomena that synaptic inputs onto developing ABNs within different dendritic compartments formed in a sequential manner (*Kelsch et al., 2008*), with the formation of centrifugal inputs preceding local dendrodendritic inputs (*Whitman and Greer, 2007*). It has been shown that the survival rate of ABNs is sensitive to sensory

experiences, which then are reflected by neuronal activities. Therefore, we postulate that ABNs that are strongly activated by centrifugal inputs may have a higher chance to survive, which can explain the state-dependent requirement of ABNs for MC suppressive responses that we have observed.

The activity of inhibitory circuits has been shown to be sensitive to behavioral states in various brain areas. Intracellular recordings from excitatory neurons in the primary visual cortex revealed that inhibitory inputs are more prevalent in the awake state than in anesthesia (*Haider et al., 2013*). In the primary auditory cortex, task engagement suppresses sound-evoked responses and sharpens the tuning of excitatory neurons (*Lee and Middlebrooks, 2011*; *Otazu et al., 2009*). This effect is mediated by subtype-specific modulation of local inhibitory neurons (*Kuchibhotla et al., 2017*). In the olfactory bulb, anesthesia suppresses local inhibitory neurons (*Kato et al., 2012*; *Wachowiak et al., 2013*). Together with the current study, these results suggest that state-dependent engagement of inhibitory circuits and suppression of excitatory responses may be a common principle conserved across brain areas.

We also note that task engagement does not only affect suppressive responses of MCs. In passive exposure, excitatory responses were also reduced compared to the task condition, although this effect was insensitive to ABN ablation (*Figure 7—figure supplement 1*). Therefore, it appears that there are additional, ABN-independent mechanisms modulating MC responses in a state-dependent manner. These probably include various feedback systems, which can modulate MC activity and olfactory behavior (*Castillo et al., 1999*; *Chaudhury et al., 2009*; *Escanilla et al., 2010*; *Kapoor et al., 2016*; *Linster et al., 2001*; *Ma and Luo, 2012*; *Nunez-Parra et al., 2013*; *Rothermel et al., 2014*). It is likely that some of the functions of these systems are independent of ABNs.

Taken together, we propose a model that task engagement increases the dynamic range of MC responses through top-down modulation from higher brain areas, which acts at least partially through young ABNs. Consistent with this notion, inactivation of piriform cortex, a main source of feedback projections to the olfactory bulb, enhances excitatory MC responses (*Otazu et al., 2015*), supporting the inhibitory role of cortical feedback. These dynamics of ABNs may underlie the observations that experience and learning profoundly shape the representations of odorant stimuli by olfactory bulb principal neurons (*Chu et al., 2016*, *2017*; *Doucette and Restrepo, 2008*; *Gschwend et al., 2015*; *Kato et al., 2012*; *Yamada et al., 2017*). Thus, the olfactory bulb functions as a dynamic, adaptive filter for incoming odorant information depending on behavioral demands, and adult neurogenesis is essential for this adaptive role of the olfactory bulb.

# Materials and methods

**Key resources table**

| Reagent type (species) or resource | Designation | Source or reference | Identifiers | Additional information |
|---|---|---|---|---|
| strain, strain background | Gfap-tk, ICR | PMID: 21814201 | | Cameron Lab |
| strain, strain background | Cdhr1-Cre, C57Bl/6 | RIKEN Brain Research Center | ID_source:RBRC02189 | |
| strain, strain background | mGfap-Cre, C57Bl/6 | PMID: 15494728 | | Sofroniew Lab |
| strain, strain background | Slc17a7-LoxP-TeNT, C57Bl/6 | PMID: 24760839 | | Imayoshi Lab |
| transfected constrct | AAV2.1 hsyn-FLEX-GCaMP6f | Upenn Vector Core | ID_source:CS1165 | |
| transfected constrct | AAV2.1-EF1a-FLEX-taCasp3-TEVp | Upenn Vector Core | ID_source:V3734TI-S | |
| transfected constrct | AAV2.1-CMV-PI-Cre-rBG | Upenn Vector Core | ID_source:CS1235 | |
| antibody | rat anti-BrdU | AbD serotec | ID_source:OBT0030 | Dilution: 1:500 |
| antibody | goat anti-rat Alexa 488 | Thermo Fisher | ID_source:RRID:AB_2534074 | Dilution: 1:1000 |
| antibody | goat anti-Doublecortin | Santa Cruz | ID_source:SC8066 | Dilution: 1:400 |
| antibody | donkey anti-goat Alexa 488 | Thermo Fisher | ID_source:RRID:AB_2534102 | Dilution: 1:1000 |

| Reagent type (species) or resource | Designation | Source or reference | Identifiers | Additional information |
|---|---|---|---|---|
| antibody | goat anti-GFAP | Santa Cruz | ID_source:SC6170 | Dilution: 1:400 |
| antibody | mouse anti-NeuN | Millipore | ID_source:MAB377 | Dilution: 1:400 |
| antibody | rabbit anti-VAMP2 | Synaptic Systems | ID_source:104_202 | Dilution: 1:500 |
| antibody | DAPI | Vector Labs | ID_source:H1200 | |
| chemical compound, drug | Valganciclovir | Genentech | ID_source:NDC 0004-0039-09 | |
| chemical compound, drug | heptanal | Sigma | ID_source:111-71-7 | |
| chemical compound, drug | ethyl-tiglate | Sigma | ID_source:5837-78-5 | |
| software, algorithm | Matlab | https://www.mathworks.com/products/matlab.html | ID_source:RRID:SCR_001622 | |

## Subjects

All procedures were in accordance with protocols approved by the Institutional Animal Care and Use Committee at UCSD or Kyoto University and guidelines of the National Institute of Health.

For all experiments, mice were housed in plastic cages with standard bedding in a room with a reversed light cycle (12 hr-12hr), and all experiments were performed during the dark period.

All experiments except suppression of hippocampal postnatally-born neurons were performed at UCSD. *Gfap-tk* mice were generous gifts from H. Cameron with ICR background. *Cdhr1-Cre* mice were originally acquired from RIKEN Brain Research Center and backcrossed at least four generations to C57Bl/6. Only male mice were used. All littermates were used for experiments, roughly 50% of which were positive for *Gfap-tk*, and the mice negative for *Gfap-tk* served as control. The experimenters were blinded to the genotype of each mouse until the end of the experiments. The genotypes were confirmed by both PCR and post hoc Doublecortin immunostaining, which were always consistent with each other (PCR negative mice always showed Doublecortin signals and vice versa).

Suppression of hippocampal postnatally-born neurons was performed at Kyoto University. *mGfap-Cre* mice (*Garcia et al., 2004*) were crossed with *Slc17a7-LSL-TeNT* mice (*Sakamoto et al., 2014*). Both strains were maintained on the C57Bl/6 background. The experimenters were blind to the genotype of each mouse during the experiments, after which double transgenic mice were identified by PCR. *Slc17a7-LSL-TeNT* single transgenic mice served as control. No behavioral abnormalities were observed in the *mGfap-Cre* and *Slc17a7-LSL-TeNT* single transgenic mice. All behavioral tests were carried out with 3-months-old male mice.

## Valganciclovir (VGCC) treatment

VGCC (Genentech) was dissolved in drinking water at 0.63 mg/ml before water restriction, and mixed with powdered food (Harlan, Indianapolis, IN) at 0.44 mg/g during water restriction, to achieve approximately 0.1 mg/g body weight/day. Mice were 10–12 weeks old at the beginning of VGCC treatment.

## Surgeries

After 6 weeks of continuous VGCC treatment, mice were anesthetized with isoflurane (3% induction, 0.7–2% maintenance) and surgeries were performed as previously described (*Kato et al., 2012*). Briefly, a stainless-steel custom headplate was secured onto the skull with cyanoacrylate glue, and an optical glass window (1 × 2 mm, oval) was implanted above the right olfactory bulb craniotomy and was secured by dental cement.

## Viral injection

To express GCaMP6f in mitral cells, a viral vector containing a Cre-dependent, GCaMP6f-expressing construct (AAV2.1 hsyn-FLEX-GCaMP6f, UPenn Vector Core, 1:11 diluted in saline) was injected into the craniotomy (20 nl / site, four sites, 250 µm depth).

To ablate a random subset of cells in GCL, a mixture of viruses containing Cre-expressing construct (AAV2.1-CMV-PI-Cre-rBG, UPenn Vector Core, 1:10 dilution in saline) and Cre-dependent modified

Caspase3 (*Yang et al., 2013*), AAV2.1-EF1a-FLEX-taCasp3-TEVp, custom prep by UPenn Vector Core, 1:1 dilution in saline) was injected into the olfactory bulb (300 nl or 500 nl, one site, 0.75 mm M-L, 0.8 mm anterior from the inferior cerebral vein, 1.5 mm D-V, injection speed: 100 nl / min) through a small craniotomy. For all behavioral experiments and a subset of histology experiments, the injections were bilateral. For the other histology experiments, the injections were unilateral and the uninjected hemisphere served as control.

## BrdU treatment

To validate the effectiveness of adult neurogenesis ablation, after 6 weeks of continuous VGCC treatment, mice (six control, six ablation) were treated with BrdU for three consecutive days, and were sacrificed 7 days later for immunostaining. BrdU powder was dissolved in drinking water at 1 mg/ml to achieve approximately 0.2 mg/g body weight/day.

## Immunostaining and cell counting

30 μm-thick olfactory bulb coronal sections were prepared with a microtome (Thermo Fisher) and mounted on pre-coated slides. Immunostaining was then performed with overnight primary antibody and 2 hr secondary antibody incubation. For BrdU staining, sections were incubated at 37°C in HCl (6% in water) for 30 min, and neutralized by borate acid buffer (0.5 M) for 10 min prior to incubation with the primary antibody. Both primary and secondary antibodies were diluted in blocking buffer (0.3% TritonX-100, 1% serum from the same species as secondary antibody, 0.1% bovine serum albumin, 0.1 M ph7.4 PBS). BrdU: primary (rat, AbD serotec, Oxford, UK, RRID: AB_10015293), 1:500, secondary (goat, Alexa 488, Thermo Fisher, Waltham, MA, RRID: AB_2534074), 1:1000. Doublecortin: primary (goat, Santa Cruz, Dallas, TX), 1:400, secondary (donkey, Alexa 488, Thermo Fisher, Waltham, MA, RRID: AB_2534102), 1:1000. GFAP: primary (goat, Santa Cruz, Dallas, TX), 1:400, secondary (same as doublecortin). NeuN: primary (mouse, Millipore, Temecula, CA), 1:400, secondary (goat, Alexa 488, Thermo Fisher, Waltham, MA, RRID: AB_2633275), 1:1000. DAPI: 1:10,000 (Invitrogen, Carlsbad, CA) for *Figure 1C,D,E*, and Vectashield mounting medium (Vector Labs, Burlingame, CA) for *Figure 3C,E*.

GFAP, BrdU, NeuN and DAPI quantification was performed manually using ImageJ. Representative sections (~4 for each animal) were chosen, and in each section, four rectangle areas were selected for counting, each encompassing the entire depth of the GC layer from dorsal, ventral, medial and lateral sides where signals were relatively homogenous. For GFAP signals, only complete structures containing soma were counted. For BrdU, all clearly visible puncta were included. To measure GCL width, 3–4 coronal sections from the widest segment of each OB were selected, and the distances between the central line of ventricle to the mitral cell layer on both medial and lateral sides were measured using ImageJ, and then averaged.

## Odorant delivery

Odorants (Sigma) were diluted in mineral oil (Thermo Fisher, Waltham, MA) to a calculated vapor pressure of 200 ppm. A custom-built olfactometer mixed saturated odorant vapor 1:1 with filtered, humidified air for a final concentration of 100 ppm. Air flow rate was controlled at 1 L / min by a mass flow controller (Aalborg, Orangeburg, NY). Heptanal and Ethyl-tiglate were selected based on their structural dissimilarity and strong odorant-evoked responses in dorsal olfactory bulb.

## Behavior

Water restriction started ~1 week after surgery and 14–18 days prior to the start of behavioral training. Mice were given at least 1 ml of water per day to maintain the body weight ≥80% of the initial value. The behavioral program was controlled by a real-time system (C. Brody). Two lick ports with infrared beam detector were available for left and right licks. A correct trial (determined by the first lick during the answer period) was rewarded with ~6 μl of water. Each daily training session consisted of 150 trials unless mice disengaged earlier.

## Pre-training

In the first session, mice were rewarded for both left and right licks during a 2 s answer time. The inter-trial interval (ITI) was increased from 1 s to 3 s. In the second session, 80% Heptanal 20% Ethyl-tiglate (80H20E) mixture was delivered for 4 s in each trial, followed by a 2 s answer period during which a left lick was rewarded. Right lick during the answer period would terminate the trial without reward or punishment. ITI was increased by 3 s every ~20 trials up to 15 s and was fixed at 15 s for all the following sessions. In the following session, 20% Heptanal 80% Ethyl-tiglate (20H80E) mixture was delivered in each trial to train right licks.

## Easy discrimination

Once mice could perform correctly for >90% of 60 consecutive trials in both the left- and right-lick pre-training sessions, we began the easy discrimination task in which 80H20E and 20H80E were pseudo-randomly delivered in each trial with no more than three successive trials of the same mixture. 80H20E and 20H80E signaled left and right lick trials, respectively. Incorrect responses terminated the trials without reward or punishment. Mice were trained with this easy discrimination task until they achieved >80% success rate in an entire session.

## Difficult discrimination

In each trial, one of the eight mixtures (left lick: 54H46E, 52H48E, 51.5H48.5E and 51H49E; right lick: 49H51E, 48.5H51.5E, 48H52E, 46H54E) was pseudo-randomly delivered so that no consecutive trials were of the same mixture and each mixture was delivered at about the same frequency. Mice were trained with this difficult discrimination task for 10 sessions.

## Passive exposure

A separate cohort of mice went through a passive experience paradigm, where they experienced the same odorants through the same timeline (pre-training, easy discrimination to difficult discrimination) with the same trial structure and session duration (150 trials) passively without task engagement. The number of easy discrimination sessions (4) was determined based on the median of session numbers during the task engagement experiment.

## Open field test

An open field test was performed on a subset of mice who had completed the behavioral training. An enclosed cubic box (edge: 40 cm) made with black acrylic boards was used as the open field. Each mouse was placed in the center of the box floor, and was allowed to explore freely for 5 min. An infrared camera (29 frames/s) was secured at the center of the box ceiling to record the location of the mouse. Speed, distance and location were analyzed on a frame-by-frame basis using custom code in MATLAB (RRID: SCR_001622). The center area was defined as the 20 × 20 cm area in the center of the floor.

## Image acquisition

Two-photon imaging was performed with a commercial microscope (B-scope, Thorlabs, Newton, NJ) with 925 nm laser excitation (Mai-Tai, Spectra-physics, Santa Clara, CA) at the frame rate of 26–28 Hz. Each frame was 512 × 512 pixels with the average field of view of 546 × 467 μm. Imaging was performed continuously within each of 4000-frame (~44 s) segments, which were separated by a 6 s inter-segment interval. Trials that overlapped with these intervals were discarded. The average image from the first imaging session was used as a template to identify the same imaging field in the following sessions.

## Data analysis

The image time series were first processed for full-frame motion correction with a custom program in MATLAB.

### ROIs

Regions of Interest (ROIs) were manually drawn around each mitral cell with a custom MATLAB program on the average image of each session. ROIs were added or removed by comparing across all imaging sessions to make sure all analyzed cells were visible and appeared healthy in every session. A background ROI was also manually drawn in an area adjacent to each cell body ROI without cellular structures. The values of the pixels within each cell body and background ROI were averaged to generate two fluorescence time series (F). For each trial, (F(background) - mean(F(background))) was subtracted from F(cell body) to derive the final cell activity trace. The 5 s period before odorant onset was used as baseline for each trial and the activity trace for each trial was normalized to the mean of the baseline period to calculate $F/F_0$ and dF/F. The total number of mitral cells and mice imaged were: control: 703 cells in 12 mice; ablation: 540 cells in 10 mice; control passive: 416 cells in 10 mice; ablation passive: 298 cells in seven mice.

### Defining responsive cells

Responsive mitral cells were defined in each session as previously described (*Chu et al., 2016*) using trial traces smoothed with MATLAB 'smooth' function (smooth factor = 6). A mitral cell was classified as responsive to a given odorant mixture if both of the following criteria were met:

### Criterion 1

$F/F_0$ is significantly different by Wilcoxon rank sum test ($p < 0.05$) between each time point (frame) of all trials and baseline frames of all trials for at least 75% of the time points within any 0.5 s time window during the 4 s odorant period.

### Criterion 2

The difference between trial-averaged $F/F_0$ and the grand average of baseline frames of all trials exceeds 0.20 in at least one frame during the 0.5 s window that meets Criterion 1.

### Decoder analysis

Mitral cell population response in each trial was expressed as a population activity vector by averaging $F/F_0$ values 0–2 s and 2–4 s of odorant period for each cell and concatenating these two values across all cells. For each mouse, every decoding process ran 100 iterations. For decoding using all mitral cells, 20 cells were randomly selected from all cells in each iteration. This number (20 cells) was decided based on the mouse with the smallest number of imaged mitral cells. *'8-odorant decoding'*: in each iteration, centroids for all eight odorants were calculated by averaging activity vectors of all trials for each respective odorant excluding the test trial. Euclidian distances between the test trial and all centroids were calculated using the MATLAB function 'pdist', and the centroid with the shortest distance defined the decoded odorant. If the decoded odorant matches the actual odorant delivered for that trial, the trial was considered correctly decoded. The final decoder accuracy of each mouse was the result of averaging fraction of correctly decoded trials across 100 iterations. *'Pairwise decoding' (Figure 4G):* the same decoder analysis was also performed in a pairwise fashion for every pair of the eight odorants in the difficult discrimination.

### Calculating correlation coefficients

To calculate correlation coefficients between behavioral performance, decoder accuracy and response level described in *Figures 4H and 6*, the MATLAB function 'corrcoef' was used.

### Linear regression model

To test whether the effect of task engagement on mitral cell responses is different between control vs. ablation animals, matrices (number of mice × number of sessions) of responsive fractions for control task, ablation task, control passive and ablation passive were constructed and fit with the linear model: *Responsive fraction = a × genotype (control/ablation) b × condition (task/passive) c × session d × genotype×condition e* using Matlab function 'fitlm'.

## Acknowledgements

We thank AN Kim, K O'Neil, T Loveland, O Arroyo and L Hall for technical assistance, and S Ackerman, N Spitzer and members of the Komiyama lab, especially R Hattori, E Hwang and A Ramot for discussions. This research was supported by grants from NIH (R01 DC014690-01, R21 DC012641, R01 NS091010A, U01 NS094342, R01 EY025349 and P30EY022589), New York Stem Cell Foundation, David and Lucile Packard Foundation, Pew Charitable Trusts, McKnight Foundation, Kavli Institute for Brain and Mind, and NSF (1734940) to TK, from Japan Science and Technology Agency (PRESTO) and Human Frontier Science Program to TK and II, and from the Ministry of Education, Culture, Sports, Science and the Technology of Japan (15H05570, 16H06529) to II.

## Additional information

### Funding

| Funder | Grant reference number | Author |
| --- | --- | --- |
| National Institute on Deafness and Other Communication Disorders | R01 DC014690-01 | Takaki Komiyama |
| National Institute of Neurological Disorders and Stroke | R01 NS091010A | Takaki Komiyama |
| National Eye Institute | R01 EY025349 | Takaki Komiyama |
| New York Stem Cell Foundation | | Takaki Komiyama |
| David and Lucile Packard Foundation | | Takaki Komiyama |
| Pew Charitable Trusts | | Takaki Komiyama |
| McKnight Foundation | | Takaki Komiyama |
| Kavli Foundation | | Takaki Komiyama |
| National Science Foundation | 1734940 | Takaki Komiyama |
| Human Frontier Science Program | | Itaru Imayoshi Takaki Komiyama |
| Ministry of Education, Culture, Sports, Science, and Technology | 16H06529 | Itaru Imayoshi |
| Japan Science and Technology Agency | | Itaru Imayoshi Takaki Komiyama |
| National Institute on Deafness and Other Communication Disorders | R21 DC012641 | Takaki Komiyama |
| National Institute on Deafness and Other Communication Disorders | U01 NS094342 | Takaki Komiyama |
| National Eye Institute | P30EY022589 | Takaki Komiyama |
| Ministry of Education, Culture, Sports, Science, and Technology | 15H05570 | Itaru Imayoshi |

The funders had no role in study design, data collection and interpretation, or the decision to submit the work for publication.

## Author contributions

Wankun L Li, Data curation, Formal analysis, Investigation, Visualization, Methodology, Writing – original draft, Writing – review and editing; Monica W Chu, Data curation, Formal analysis, Investigation, Methodology; An Wu, Formal analysis, Investigation, Writing – review and editing; Yusuke Suzuki, Formal analysis, Investigation; Itaru Imayoshi, Data curation, Investigation, Methodology, Writing – review and editing, Conceptualization, Supervision, Funding acquisition, Project administration; Takaki Komiyama, Methodology, Writing – original draft, Writing – review and editing, Conceptualization, Supervision, Funding acquisition, Project administration, Resources

## Author ORCIDs

Wankun L Li http://orcid.org/0000-0002-2809-039X
An Wu http://orcid.org/0000-0002-6593-3181
Takaki Komiyama http://orcid.org/0000-0001-9609-4600

## Ethics

All procedures were in accordance with protocols approved by the Institutional Animal Care and Use Committee at UCSD (protocol number s10221) or Kyoto University (permit number Med Kyo 16216) and guidelines of the National Institute of Health.

## Decision letter and Author response

Decision letter https://doi.dx.org/10.7554/eLife.33006.016
Author response https://doi.dx.org/10.7554/eLife.33006.017

---

# Additional files

## Supplementary files

• Transparent reporting form.
DOI: https://doi.org/10.7554/eLife.33006.015

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
