## [Decision Letter]

Thank you for submitting your article "Adult-born neurons facilitate olfactory bulb pattern separation during task engagement" for consideration by*eLife*. Your article has been favorably evaluated by Gary Westbrook (Senior Editor) and three reviewers, one of whom, Naoshige Uchida (Reviewer #1), is a member of our Board of Reviewing Editors. The following individual involved in review of your submission has agreed to reveal their identity: Florin Albeanu (Reviewer #2). The reviewers have discussed the reviews with one another and the Reviewing Editor has drafted this decision to help you prepare a revised submission.

Summary:

Li and colleagues examined the role of adult neurogenesis in olfactory discrimination. The authors used transgenic mice expressing herpes simplex virus thymidine kinase (TK) under control of the glial fibrillary acidic protein (GFAP) promotor (GFAP-TK mice). TK renders mitotic cells sensitive to the antiviral drug (Valganciclovir, VGCC) while post-mitotic cells remained intact. This allowed the authors to ablate adult born neurons (ABNs) in an inducible manner while keeping other cells, such as glial cell which expresses GFAP, remained unaffected (Snyder et al., 2011). Indeed, the authors were able to suppress ABNs in the olfactory bulb (OB) almost completely.

Using this method, the authors first demonstrate that ABN ablation impaired the mice's performance in fine, but not coarse, discrimination. Interestingly, ABN ablations in the hippocampus or ablation of a similar or even greater number of random OB neurons did not affect odor discrimination performance. To link changes in neuronal responses in the OB and behavioral performance, the authors then imaged calcium signals from mitral cells (MCs) in the OB during odor discrimination performance. The authors' analysis shows that inhibitory responses of mitral cells were reduced in ABN ablation mice, and the extent of this reduction was correlated with the discriminability of population neuronal activities and the behavioral impairment in fine odor discrimination. These results together support the role of ABN neurons in refining odor representation in the OB, predominantly shaping inhibitory responses in mitral cells. The authors further present the results indicating that the requirement for ABNs is context-dependent – the relevant context is whether the animals are engaged in the task or not.

It has been proposed that adult neurogenesis plays important roles in olfactory functions, but the exact role and the mechanism underlying it remains debated and largely unknown. This study addresses these important questions, and provides very interesting results. The experiments combine a novel technique of cell ablation, two-photon calcium imaging and behavior. The results are presented and discussed in a careful manner. The manuscript was well-written and easy to follow. Together this study advances our understanding of the importance of adult neurogenesis in an olfactory function, and its neural mechanism. Although all the reviewers were very positive about the study, they raised a number of concerns and suggestions, to which we would like to see your response.

Essential revisions:

1) The authors performed a control experiment to ablate hippocampal ABNs to support the importance of ABNs in the OB, rather than hippocampus. It is important to demonstrate that this control experiment ablated ABNs in the hippocampus in a similar or a greater extent compared to the original experiment. Currently, no data was presented.

2) Can the authors show that post-mitotic, GFAP-positive glial cells are not reduced by the manipulation?

3) If the authors have imaged from mitral cells in animals where ~20% of the random GCs were ablated, this information would be useful. Specifically, it would be important to determine what is the performance of the decoders based on the MC ensemble activity in the random ablation group compared to the control. Is there a decay in the decoder accuracy, and if so, is it substantially lower compared to the ABN ablation group?

[Editors' note: further revisions were requested prior to acceptance, as described below.]

Thank you for resubmitting your work entitled "Adult-born neurons facilitate olfactory bulb pattern separation during task engagement" for further consideration at*eLife*. Your revised article has been favorably evaluated by Gary Westbrook (Senior Editor) and three reviewers, one of whom is a member of our Board of Reviewing Editors. The authors have addressed most of the previous concerns, and the manuscript has improved. However, the reviewers pointed out some remaining issues that should be addressed before publication as follows:

1) The ablation method in the main experiment using gfap-TK ablates adult-born neurons (ABNs) not only in the olfactory bulb (OB), but also in the hippocampus. It is critical for the authors to show that the deficit in fine odor discrimination results from ABN ablation in the OB and not the hippocampus. The authors performed a new experiment to address this by inactivating neurons in the hippocampus with tetanus toxin (TeNT) in a Cre-dependent manner in*mGfap-Cre*mice using a transgenic approach. This approach means that inactivation is not specific to ABNs but includes all postnatally-born neurons. The authors should compare the extent of ablation/inactivation focusing on ABNs, for example, by using BrdU labeling. In the OB, they were able to ablate almost all ABNs using gfap-TK. If this is true in the hippocampus, it appears very difficult to achieve comparable inactivation through their TeNT approach.

This raises two issues. First, the characterization of the control experiments without focusing on ABNs is problematic. Second, there may be a fundamental technical difficulty in achieving near complete inactivation of ABNs in the hippocampus (i.e. performing an ideal control experiment) using an available technique. If the latter is true, a viable approach would be to clarify the limitation of the control experiments explicitly in text, and carefully interpret the results. The reviewers agree that, given the significance of main results, this issue should not prevent the authors from publishing this work, provided that the authors discuss the limitation of the control experiments that they performed.

Additionally, please explain the method of this control experiment more explicitly. For instance, please spell out and explain 'LSL' and 'TeNT'. Currently, it is difficult to know what was done just by reading the main text.

2) The authors show that the discriminability of odor-evoked activities in the OB correlates with behavioral performance in fine odor discrimination. This is a very interesting result and one of the core results. The reviewers previously asked the authors to perform a similar experiment in a control condition where they ablated a random population of OB neurons. Despite the authors' effort to collect this data, their experiment did not go well. The reviewers noted that this is still a salient point, and we request that the authors discuss that this issue needs to be addressed in the future.

3) Although the behavioral performance keeps increasing during training, decoder accuracy does not improve. Izumi points out some potential discrepancy with their previous work (Chu et al., 2016). It is indeed possible that behavioral training induces changes in other parts of the brain, which may account for the improvement in performance during the training period. However, in Chu et al. 2016, there seems to be a clear distinction in the OB representation between correct rejection trials vs. false alarm trials. Why do they not see a consequence of behavioral improvement here? Please clarify this point.

---

## [Author Response]

[…] It has been proposed that adult neurogenesis plays important roles in olfactory functions, but the exact role and the mechanism underlying it remains debated and largely unknown. This study addresses these important questions, and provides very interesting results. The experiments combine a novel technique of cell ablation, two-photon calcium imaging and behavior. The results are presented and discussed in a careful manner. The manuscript was well-written and easy to follow. Together this study advances our understanding of the importance of adult neurogenesis in an olfactory function, and its neural mechanism. Although all the reviewers were very positive about the study, they raised a number of concerns and suggestions, to which we would like to see your response.

We are very grateful to the reviewers for their enthusiasm and constructive comments about the significance of our findings, experimental design, and writing. We have carefully considered their comments and revised our manuscript with new analyses and experiments as detailed below.

First, we would like to include a new behavioral experiment as an additional control for impairment of fine discrimination in the ABN ablation animals. Although this issue was not raised by the reviewers, we have been asked in other discussions whether the deficits in the difficult 8-odorant discrimination task were due to the similarity of odorant mixtures or due to the complexity of having 8 different odorant mixtures. To address this issue, we trained another cohort of ablation and control animals with an ‘easy 8-odorant discrimination task’ with 8 mixtures of very different ratios. Both control and ablation animals performed as well in this task as in the easy 2-odorant discrimination task, establishing that the impairment of the ablation animals in the difficult 8-odorant task was due to the similarity of mixtures (Figure 2—figure supplement 1).

Essential revisions:1) The authors performed a control experiment to ablate hippocampal ABNs to support the importance of ABNs in the OB, rather than hippocampus. It is important to demonstrate that this control experiment ablated ABNs in the hippocampus in a similar or a greater extent compared to the original experiment. Currently, no data was presented.

The approach used for the hippocampal suppression experiment targeted TeNT to all postnatal-born neurons in the dentate gyrus (Sakamoto et al., 2014), of which ABNs are a subset. (We erroneously referred to this experiment as ‘hippocampal ABN suppression’ in the initial submission. We apologize for the error and now we refer to this as ‘suppression of hippocampal postnatally-born neurons’ in the revised manuscript.) We performed new experiments to quantify the degree of suppression and found that this method targets ~35% of neurons in dentate gyrus, as opposed to estimated ~5% in the case of ABN ablation. So this control experiment affected a larger fraction of neurons in dentate gyrus than ABN ablation GFAP-TK. We have included this information inFigure 1—figure supplement 1.

2) Can the authors show that post-mitotic, GFAP-positive glial cells are not reduced by the manipulation?

We have shown that the density of GFAP-positive glial cells was not affected by GFAP-TK ablation in the olfactory bulb granule cell layer. Please refer toFigure 1E. A previous publication also confirmed this in dentate gyrus (Snyder et al., 2011). We have reproduced these results and now include the data inFigure 1—figure supplement 1.

3) If the authors have imaged from mitral cells in animals where ~20% of the random GCs were ablated, this information would be useful. Specifically, it would be important to determine what is the performance of the decoders based on the MC ensemble activity in the random ablation group compared to the control. Is there a decay in the decoder accuracy, and if so, is it substantially lower compared to the ABN ablation group?

We did not have existing data to address this point, so we made an attempt to perform the requested experiment by ablating random GCs bilaterally and labeling MCs in the right OB with GCaMP6f. We attempted this in Pcdh21-Cre mice with unilateral injection of AAV2/1-FLEX-GCaMP6f in the MC layer and bilateral injections of a combination of AAV2/1-Cre and AAV2/1-FLEX-tCasp3 in the center of the GC layer.

First, we tried this by one surgery of the bilateral injections of the Cre-Casp combination, followed 3-5 days later by another surgery for unilateral craniotomy and MC injection. We were not able to achieve craniotomies of quality sufficient for imaging, presumably due to damage from the first injections (n = 3 mice).

Next, we performed all injections and craniotomy within one surgical session (n = 10 mice). Unfortunately, 7 out of 10 mice died within 5 days post-op possibly due to the long time required for the surgery. Of the remaining 3 animals, the craniotomy quality of 1 animal was too poor for imaging. We are currently performing the experiment with the remaining two animals, but the data will not be sufficient to draw any concrete conclusions.

We will continue to perform these experiments. However, despite great efforts, we were not able to complete these experiments within the time frame of the revision of the current manuscript, for which we apologize.

[Editors' note: further revisions were requested prior to acceptance, as described below.]

1) The ablation method in the main experiment using gfap-TK ablates adult-born neurons (ABNs) not only in the olfactory bulb (OB), but also in the hippocampus. It is critical for the authors to show that the deficit in fine odor discrimination results from ABN ablation in the OB and not the hippocampus. The authors performed a new experiment to address this by inactivating neurons in the hippocampus with tetanus toxin (TeNT) in a Cre-dependent manner in mGfap-Cre mice using a transgenic approach. This approach means that inactivation is not specific to ABNs but includes all postnatally-born neurons. The authors should compare the extent of ablation/inactivation focusing on ABNs, for example, by using BrdU labeling. In the OB, they were able to ablate almost all ABNs using gfap-TK. If this is true in the hippocampus, it appears very difficult to achieve comparable inactivation through their TeNT approach.This raises two issues. First, the characterization of the control experiments without focusing on ABNs is problematic. Second, there may be a fundamental technical difficulty in achieving near complete inactivation of ABNs in the hippocampus (i.e. performing an ideal control experiment) using an available technique. If the latter is true, a viable approach would be to clarify the limitation of the control experiments explicitly in text, and carefully interpret the results. The reviewers agree that, given the significance of main results, this issue should not prevent the authors from publishing this work, provided that the authors discuss the limitation of the control experiments that they performed.Additionally, please explain the method of this control experiment more explicitly. For instance, please spell out and explain 'LSL' and 'TeNT'. Currently, it is difficult to know what was done just by reading the main text.

We apologize for our lack of clarity on this new experiment. We also thank the reviewers for appreciating the difficulty of making direct comparisons of the two different methods that were available to us and for inviting us to discuss the limitation of the control experiments. We reiterate that our histology results shown inFigure 3—figure supplement 1Aand 1B demonstrate that this strategy, which should target all postnatally-born neurons, indeed targeted a very large fraction (more than one third) of DG granule cells. We also now provide new data (Figure 3—figure supplement 1C) that shows that the immunostaining signal for VAMP2 in the mossy fiber region of CA3 (i.e. axon terminals of DG granule cells) exhibits a marked decrease in the TeNT-expressing animals, providing support for functional effects of TeNT expression in suppressing the output of postnatally-born DG granule cells. The remaining VAMP2 signal probably originates from the spared, developmentally-born DG granule cells.

We believe that these results provide strong support that our control experiments suppressed the output of most, if not all, DG ABNs (and other postnatally-born neurons that were born earlier). However, we concede that this is not directly comparable to the GFAP-TK experiments and included the following sentence in the main text: “We do note, however, that a formal possibility remains that a small fraction of DG ABNs that may have been spared in the ‘DG suppression’ method may partially contribute to the behavioral deficits of the*Gfap-tk*^+^ ablation group.”

Following the reviewers’ suggestions, we have expanded the explanation of this new experiment in the main text, including spelling out LSL (loxP-STOP-loxP) and TeNT (tetanus neurotoxin) and explaining how this method targets postnatally-born excitatory neurons and thus spares OB granule cells.

2) The authors show that the discriminability of odor-evoked activities in the OB correlates with behavioral performance in fine odor discrimination. This is a very interesting result and one of the core results. The reviewers previously asked the authors to perform a similar experiment in a control condition where they ablated a random population of OB neurons. Despite the authors' effort to collect this data, their experiment did not go well. The reviewers noted that this is still a salient point, and we request that the authors discuss that this issue needs to be addressed in the future.

We agree with the reviewers and regret that this experiment turned out to be very difficult. Following their suggestion, we have included the following paragraph in Discussion: “A prediction of our model is that MC suppressive responses and discriminability would be less affected (more similar to control animals) in the random ablation animals than in the ABN ablation animals. […] Therefore this remains a question to be addressed in the future.”

3) Although the behavioral performance keeps increasing during training, decoder accuracy does not improve. Izumi points out some potential discrepancy with their previous work (Chu et al., 2016). It is indeed possible that behavioral training induces changes in other parts of the brain, which may account for the improvement in performance during the training period. However, in Chu et al. 2016, there seems to be a clear distinction in the OB representation between correct rejection trials vs. false alarm trials. Why do they not see a consequence of behavioral improvement here? Please clarify this point.

We agree that there are important points to be discussed here and thank the reviewers for encouraging us to address them. We believe that there are cases in which the discriminability of MC responses is the limiting factor for behavioral performance, when MC decoder accuracy correlates with the behavioral performance. However, in our current task, this does not seem to be the case, as behavior can improve without improvement in MC decoder accuracy. This is presumably because MC discriminability had already been optimized during the easy discrimination task, and the subsequent learning is mainly limited by sensory-motor association learning performed by downstream areas. We intend to perform further experiments to clarify these scenarios in the future. For the current manuscript, we have extended the discussion on these issues by adding the following statements: “Another implication of these results is that there are likely multiple circuit origins of behavioral errors. […] However, in the current task, the downstream areas (e.g. olfactory cortex) and their ability to associate MC outputs with behavioral choice may be a main source of behavioral variability.”